

Atmospheric
Measurement
Techniques

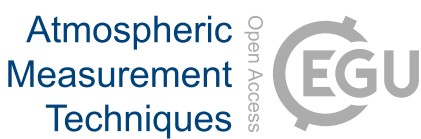

# Novel approach to observing system simulation experiments improves information gain of surface–atmosphere field measurements

**Stefan Metzger**[1,2], **David Durden**[1], **Sreenath Paleri**[2], **Matthias Sühring**[3], **Brian J. Butterworth**[2], **Christopher Florian**[1], **Matthias Mauder**[4], **David M. Plummer**[5], **Luise Wanner**[4], **Ke Xu**[6], and **Ankur R. Desai**[2]

[1]Battelle, National Ecological Observatory Network, 1685 38th Street, Boulder, CO 80301, USA
[2]Dept. of Atmospheric and Oceanic Sciences, University of Wisconsin–Madison,
1225 West Dayton Street, Madison, WI 53706, USA
[3]Institute of Meteorology and Climatology, Leibniz University Hannover, Herrenhäuser Straße 2, 30419 Hanover, Germany
[4]Institute of Meteorology and Climate Research – Atmospheric Environmental Research, Karlsruhe Institute of Technology,
Kreuzeckbahnstraße 19, 82467 Garmisch-Partenkirchen, Germany
[5]Dept. of Atmospheric Science, University of Wyoming–Laramie, 1000 E. University Ave., Laramie, WY 82071, USA
[6]Dept. of Climate and Space Sciences and Engineering, University of Michigan–Ann Arbor,
2455 Hayward St, Ann Arbor, MI 48109, USA

**Correspondence:** Stefan Metzger (smetzger@battelleecology.org)

**Abstract.** TS1 The observing system design of multidisciplinary field measurements involves a variety of considerations on logistics, safety, and science objectives. Typically, this is done based on investigator intuition and designs of prior field measurements. However, there is potential for considerable increases in efficiency, safety, and scientific success by integrating numerical simulations in the design process. Here, we present a novel numerical simulation–environmental response function (NS–ERF) approach to observing system simulation experiments that aids surface–atmosphere synthesis at the interface of mesoscale and microscale meteorology. In a case study we demonstrate application of the NS–ERF approach to optimize the Chequamegon Heterogeneous Ecosystem Energy-balance Study Enabled by a High-density Extensive Array of Detectors 2019 (CHEESEHEAD19).

During CHEESEHEAD19 pre-field simulation experiments, we considered the placement of 20 eddy covariance flux towers, operations for 72 h of low-altitude flux aircraft measurements, and integration of various remote sensing data products. A 2 h high-resolution large eddy simulation created a cloud-free virtual atmosphere for surface and meteorological conditions characteristic of the field campaign

domain and period. To explore two specific design hypotheses we super-sampled this virtual atmosphere as observed by 13 different yet simultaneous observing system designs consisting of virtual ground, airborne, and satellite observations. We then analyzed these virtual observations through ERFs to yield an optimal aircraft flight strategy for augmenting a stratified random flux tower network in combination with satellite retrievals.

We demonstrate how the novel NS–ERF approach doubled CHEESEHEAD19's potential to explore energy balance closure and spatial patterning science objectives while substantially simplifying logistics. Owing to its modular extensibility, NS–ERF lends itself to optimizing observing system designs also for natural climate solutions, emission inventory validation, urban air quality, industry leak detection, and multi-species applications, among other use cases.

**Published by Copernicus Publications on behalf of the European Geosciences Union.**

# 1   Introduction

High-quality field data are the backbone of surface–atmosphere research. However, there are inevitable trade-offs in any field measurement among cost, logistics, safety, and our ability to address science objectives. Most of the time, these trade-offs are evaluated in a heuristic or haphazard approach, or at least with limited consideration of all possible options. Nevertheless, redundancy, experience, and good fortune usually save most field measurement observing system designs (OSDs) from failure. Inspired by observing system simulation experiments (OSSEs) in the Earth system sciences (Masutani et al., 2010; Atlas et al., 2015; Hoffman and Atlas, 2016) we contemplated whether this process could be improved. In particular, we note modern advances in conducting virtual experiments within high-resolution numerical simulations (NSs) of atmospheric turbulence (e.g., Steinfeld et al., 2007). We envisioned that such NSs could yield OSSEs that help increase the information gain per funding investment, more effectively address field measurement objectives, and minimize problems that arise from safety, logistics, and cost.

Here, we derive a novel approach to OSSEs that aids surface–atmosphere synthesis at the interface of mesoscale and microscale meteorology. We then apply it to preparing field campaign resources for the Chequamegon Heterogeneous Ecosystem Energy-balance Study Enabled by a High-density Extensive Array of Detectors 2019 (CHEESE-HEAD19; Butterworth et al., 2021). At the time of this study, the CHEESEHEAD19 field measurement campaign was to be conducted in northern Wisconsin, United States of America, from June to October of 2019, with the overarching science objective of examining how the atmospheric boundary layer (ABL) responds to spatial heterogeneity in the surface–atmospheric exchanges of heat and water. Further science objectives were to test whether resulting mesoscale atmospheric processes relate to the lack of energy balance closure frequently observed by eddy covariance (EC) towers. Lastly, CHEESEHEAD19 sought to apply advanced analytics over a multiscale set of observations to yield scale-aware, energy-balanced data products that help improve model representation of subgrid processes. To that end, we wanted to harness the complementarity among various in situ and remote sensing measurement systems.

However, the joint utility of these measurement systems for addressing the science objectives was not well characterized prior to the field campaign. Moreover, their joint utility is highly sensitive to the OSD including placement of the measurements and the resulting information overlap in space and time (Fig. 1). Consequently, CHEESEHEAD19 information gain hinged on our ability to merge information among the perspectives of ground-based, airborne, and spaceborne measurements, as well as numerical models. Plentiful data that are insufficiently connected to infer meaning create the risk of data deluge rather than the next interdisciplinary

breakthrough. While advances in post-field data assimilation aim to rectify limited and variable information overlap statistically (Williams et al., 2009), only the careful OSD of the field measurements themselves can treat their root cause. We thus sought an approach that empowers making CE1 informed OSD choices for surface–atmosphere field measurements in advance of the experiment.

Simulation experiments involve asking what would happen in an imaginary scenario and trying to understand whether the predicted outcome is compatible with existing theory. This form of inquiry is not an invention of modern science, but can be traced back at least to the empirical thought experiments of ancient Greek philosophers (Palmerino, 2018; Brown and Fehige, 2019). Famously, Albert Einstein employed thought experiments for his theoretical generalizations, including in his works on special and general relativity (Norton, 1991). With the rise of NSs came the opportunity to increase the complexity and detail of thought experiments, such as how to design meteorological field measurements (e.g., Eddy, 1974; Cortina and Calaf, 2017; Gehrke et al., 2019). More frequently, however, these NSs were reserved for applications in which real-world tests would have been impractical or impossible (e.g., Wiens et al., 2003). These NSs centered on prescribing and propagating a priori knowledge, i.e., creating "data from knowledge". As a result, the findings often remained subject to strong methodological assumptions that could not necessarily be met by real-world applications. More recently, the advent of data-intensive scientific discovery promises to offset some of these limitations by providing computational facilities that aid the inference of "knowledge from data", including from artificial intelligence (Hey et al., 2009; Reichstein et al., 2019). We believe that ours is the first work that explicitly complements these two paradigms of scientific knowledge creation for deriving surface–atmosphere OSDs at the interface of mesoscale and microscale meteorology.

Previous studies employed data-intensive scientific discovery for post-field OSD assessments (e.g., Montanari et al., 2012; Koffi et al., 2013; Loescher et al., 2014; Kumar et al., 2016; Chu et al., 2017; Mahecha et al., 2017; Villarreal et al., 2019). In comparison, one innovation of the approach presented here is that it provides design information prior to deploying resources in the field. To achieve this, we expanded on recent studies of atmospheric turbulence in NSs (Sühring et al., 2018; Xu et al., 2020). Specifically, we computationally simulated virtual observations over a cloud-free CHEESEHEAD19 domain in decameter and sub-second resolution for 2 h of surface and meteorological conditions characteristic of the field campaign domain and period. This "data from knowledge" idea feeds into a framework for data-intensive scientific discovery based on physics-guided environmental response functions (ERFs; Metzger et al., 2013a; Xu et al., 2017, 2018; Metzger, 2018). The resulting explicitness promises unprecedented realism and process inference in comparison to existing pre-field OSSEs that leverage

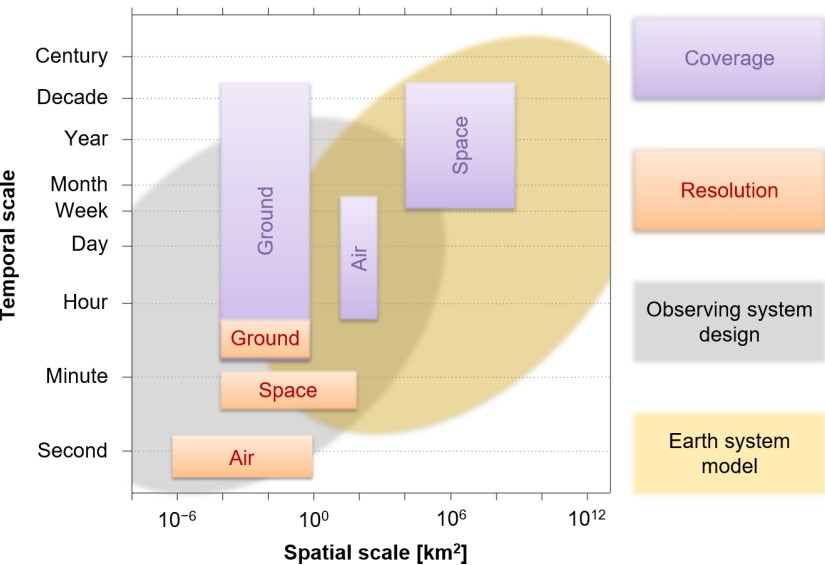

**Figure 1.** Space–time scope diagram for a surface–atmosphere synthesis observing system at the interface of mesoscale and microscale meteorology. The observing system consists of a hierarchy of ground-based (Ground), airborne (Air), and spaceborne (Space) measurements, shown in relation to two principal approaches for scaling to an information continuum: pre-field observing system design and post-field data assimilation into Earth system models. Modified after Metzger (2018).

"knowledge from data" principles (Hargrove and Hoffman, 2004; Keller et al., 2008; Zhang and Pu, 2010; Sulkava et al., 2011; Kaminski et al., 2012; Lauvaux et al., 2012; Lucas et al., 2015; Ziehn et al., 2016; Park and Kim, 2020). In the following, we derive this NS–ERF OSSE approach (in short "NS–ERF" hereafter) using the case study of designing CHEESEHEAD19 airborne flux measurements as a maiden application example. It should be noted that NS–ERF is applicable to field measurements in general, and large-scale deployments or even an aircraft operation component are by no means a requirement, which we explore further with substitution examples.

Airborne EC measurements have the particular benefit that they permit surface–atmosphere fluxes to be spatially resolved over a range of scales, from small, tower-sized flux footprints up to landscape scale. Thus, they build an important bridge among the differing scales of ground-based and spaceborne measurements (Fig. 1). Moreover, these kinds of measurements have the particular advantage that they can capture dispersive fluxes resulting from mesoscale atmospheric processes. Dispersive fluxes refer to the transport of scalar quantities by standing eddies or spatially organized time-invariant convection cells (e.g., Raupach and Shaw, 1982), which we hypothesize to be a main reason for the long-standing energy balance closure problem of tower-based eddy covariance measurements (Margairaz et al., 2020; Mauder et al., 2020). In comparison to other ground-based and spaceborne measurements, aircraft can be deployed quite flexibly in space and time. They thus provide a key to maximizing the joint information gain of harnessing complementarity among various in situ and remote sensing measurement systems. However, airborne field campaigns are also very costly and cannot be conducted continuously. Therefore, thorough planning of the flight strategy is of great importance. Previous large-scale field campaigns with similar airborne flux measurement objectives include the First ISLSCP (International Satellite Land Surface Climatology Project) Field Experiment (FIFE; Sellers et al., 1988), the Boreal Ecosystem–Atmosphere Study (BOREAS; Sellers et al., 1995), the Northern Hemisphere Climate Processes Land-Surface Experiment (NOPEX; Halldin et al., 1999), the Lindenberg Inhomogeneous Terrain – Fluxes between Atmosphere and Surface: a Long-term Study (LITFASS-98; Beyrich et al., 2002) and LITFASS-2003 (Beyrich and Mengelkamp, 2006), MAtter fluxes in Grasslands of Inner Mongolia as influenced by stocking rate (MAGIM; Butterbach-Bahl et al., 2011), ScaleX (Wolf et al., 2017), and others. In these campaigns the flight strategies were mostly based on experience and expert knowledge. Considerations included random and systematic errors (Lenschow et al., 1994) as well as the source area (or "footprint"; Schuepp et al., 1990; Kaharabata et al., 1997), sometimes supported by analyzing land cover maps using Geographic Information Systems (Metzger et al., 2013a). However, measurement errors and source areas not only depend on the flight track itself but also vary with atmospheric conditions, such as stability, wind speed, and wind direction. Moreover, focusing solely on measurement errors can be misleading in relation to heterogeneity-induced signals and result in flawed conclusions (Sühring and Raasch, 2013).

The aim of this paper is to derive the theoretical background of NS–ERF and to demonstrate its application to the

CHEESEHEAD19 OSD with a focus on the EC flight strategy. Specifically, in the following sections we test the study hypothesis that CHEESEHEAD19 information gain is sensitive to NS–ERF optimization. Two accompanying design hypotheses relate this sensitivity to the choice of flight patterns and flight sequences. Section 2 introduces CHEESE-HEAD19 and NS–ERF. Section 3 presents the NS–ERF results beginning with NS (Sect. 3.1) and ERF (Sect. 3.2) specifics, then evaluates the design hypotheses for each candidate OSD (Sect. 3.3) and provides CHEESEHEAD19 field campaign resources (Sect. 3.4). Section 4 discusses these NS–ERF results in light of the CHEESEHEAD19 OSD (Sect. 4.1), possible benefits for coordinated environmental observations in general (Sect. 4.2), and remaining challenges and future work (Sect. 4.3). Section 5 then summarizes our findings and provides an outlook.

## 2    Materials and methods

In the following we introduce CHEESEHEAD19 (Sect. 2.1) with a particular eye on general setup and science objectives, which then inform the case study realization of individual NS–ERF modules (Sect. 2.2). These include using high-resolution large eddy simulation (LES) for NS, combining virtual flux tower, aircraft, and satellite measurements in ERF, and deriving a set of NS–ERF optimality criteria that correspond to CHEESEHEAD19 science objectives. Section 2.3 further expands on this by introducing CHEESEHEAD19-specific airborne design hypotheses and candidate OSDs, and Sect. 2.4 and 2.5 detail the LES and ERF setups for this case study, respectively.

### 2.1    CHEESEHEAD19

The CHEESEHEAD19 study (Butterworth et al., 2021) included a 4-month field measurement campaign to investigate how land surface heterogeneity influences energy balance closure. The energy balance closure problem refers to the situation, common in EC measurements, whereby downward energy from radiation and ground heat flux exceeds the measured upward energy from sensible and latent heat fluxes (Foken et al., 2011; Mauder et al., 2020). Previous studies have indicated that heterogeneity is related to the energy balance closure (Stoy et al., 2013; Xu et al., 2016). The CHEESEHEAD19 project proposed to evaluate the hypothesis that mesoscale atmospheric features, driven by surface heterogeneity, are an important mechanism contributing to energy balance non-closure (Mauder et al., 2007b; Foken et al., 2011; Charuchittipan et al., 2014; Gao et al., 2016).

Due to a persistent mismatch between the scales of observations and models, the spatial and temporal scaling of surface fluxes is essential for evaluating theories on what happens within the subgrid of atmospheric models and how those feed back onto larger-scale dynamics. Therefore, an ad-

ditional science objective of the project was to use the unique multiscale set of observations to improve model representation of subgrid processes and to assess the performance of ERFs for estimating the "flux in a box" from the domain volume (Metzger, 2018; Xu et al., 2018).

The field measurement campaign was to be CE2 conducted within a $10 \times 10$ km domain of heterogeneous forest in northern Wisconsin, USA. It included patches of homogenous and mixed forests of evergreen, hardwood, and softwood deciduous trees, as well as grasses, wetlands, streams, and lakes with a characteristic surface length scale of $411 \pm 88$ m (Xu et al., 2017). The domain was relatively flat, ranging from 455 m a.s.l. in the southwest to 500 m a.s.l. in the northeast. Previous years' data from the study area showed that the summer months are characterized by light surface winds (typically $< 5$ m s$^{-1}$) coming predominately from the western hemisphere (180–360°) and daytime ABL heights ranging from 0.5 to 2.5 km above ground (mean of 1.5 km).

To measure fluxes (momentum, sensible heat, latent heat, $CO_2$) across the domain, 20 above-canopy EC towers were proposed to be deployed over a range of vegetation types (Fig. 2). They would measure all components of the energy balance, including net radiation, sensible and latent heat fluxes, and ground heat flux. The majority of the towers were expected to be instrumented at 3–32 m above ground, equaling 3–15 m above the canopy depending on land cover. The exception would be the tall tower at the center of the domain, which is an existing AmeriFlux supersite (US-PFa; Desai, 2021) that has been measuring fluxes at 30, 122, and 396 m above ground for the past 26 years (1995–2020; Desai et al., 2015).

The project also proposed to deploy a suite of remote sensing instruments (lidar, radar, sodar, ceilometers, interferometers) for measuring the mesoscale atmospheric environment (profiles of wind, $H_2O$, temperature, aerosols, ABL height). Aircraft and spaceborne remote sensing would be used to map surface characteristics that will be used for data-driven scaling methods. This would include airborne hyperspectral imaging of the land surface properties. Additionally, a land surface temperature product was planned to be developed for the domain from a multi-sensor fusion of in situ thermal drone and infrared camera imagery: ECOSTRESS, Landsat, VIIRS, and/or GOES (Desai et al., 2021).

Aircraft measurements were planned to link the differing scales of ground-based and spaceborne observations over the domain. Airborne EC fluxes (momentum, sensible heat, latent heat, $CO_2$) were measured with the University of Wyoming King Air (UWKA) during 24 research flights. The UWKA also deployed an upward-pointing cloud lidar for estimating ABL height and a downward-pointing Raman lidar for providing a three-dimensional representation of air temperature and water vapor over the domain (Wu et al., 2016). During each research flight the UWKA flew along 11 flight tracks spaced 2 km apart from each other (Fig. 2). For a given flight track the UWKA first flew outbound at

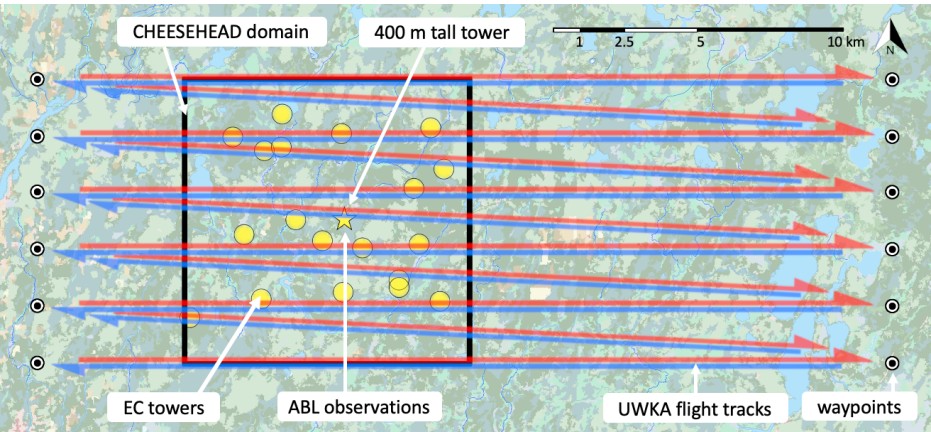

**Figure 2.** Set of candidate locations for EC towers, ABL measurements, and UWKA candidate flight tracks (west–east pattern) with respect to the 10 × 10 km CHEESEHEAD19 study domain (black box; image © Google Earth). For a given flight track the UWKA would first fly outbound at 400 m above ground (red arrows) and return at 100 m above ground (blue arrows).

400 m above ground (Fig. 2 red arrows) with an emphasis on the lidar ABL observations. This arrangement also allowed the crew to visually ensure flight safety for the immediate return at 100 m above ground (Fig. 2 blue arrows) with an emphasis on the EC surface flux observations. Owing to the CHEESEHEAD19 science objectives we will focus on the 100 m EC surface flux flights in the following sections. A more complete description of the instruments deployed during CHEESEHEAD19 can be found in Butterworth et al. (2021).

## 2.2  NS–ERF observing system simulation experiments

Virtual atmospheres emulated in NSs provide a controlled environment uniquely suited to disentangle surface–atmosphere feedbacks (e.g., Avissar and Schmidt, 1998; Kanda et al., 2004; Sühring et al., 2018; Margairaz et al., 2020; Xu et al., 2020). Our work on NS–ERF extends upon such previous applications by simulating and analyzing candidate OSDs for real-world measurements in such virtual atmospheres. NS–ERF employs value engineering principles (e.g., Park, 1998; Younker, 2003; Tohidi, 2011) to maximize the information gain on real-world measurement investments for addressing science objectives across traditional disciplinary boundaries. So long as we consider a single discipline, existing parameterizations often provide sufficient constraints to ensure meeting basic assumptions. For example, consider determining the height of an EC flux tower measurement as a function of roughness sublayer effects (e.g., Munger et al., 2012; Foken, 2017), atmospheric blending (e.g., Mason, 1988; Mahrt, 1996), and target source area (e.g., Schmid, 1997; Chen et al., 2011). However, CHEESEHEAD19 relies on harnessing complementarity across disciplines, including ground-based, airborne, and spaceborne measurements. These measurements operate on principally different space scales and timescales (Fig. 1) so that informa-

tion gain hinges on our ability to join information not only across disciplines, but also across scales. For example, the spatial context of each measurement is a function of its horizontal and vertical placement, thus providing a mechanism to maximize information overlap. Yet, optimizing each measurement's utility for joint scientific inquiry is beyond the scope of discipline-specific parameterizations. Here, we propose the extensible NS–ERF approach that explicitly simulates the joint information gain in response to different candidate OSDs for addressing user-defined design hypotheses.

Specifically, the NS–ERF approach consists of three sets of elements that interact with each other: definition elements (Fig. 3a), realization elements (Fig. 3b), and a benchmarking element (Fig. 3c). The NS–ERF sequence commences with the definition elements in Fig. 3a by defining the application objectives (i) and deriving design hypotheses (ii), OSDs (iii), and optimality criteria (iv) from them. The sequence continues to the realization elements (Fig. 3b); numerical simulations (v) create virtual measurements (vi) whose information contents are combined in a scale-aware manner using ERFs (vii). In the benchmarking element (Fig. 3c) the information gain (viii) is determined as a function of how well the ERF results (vii) for different OSDs (iii) fulfill the optimality criteria (iv). This serves as an appraisal of the design hypothesis (ii) and ultimately of the suitability of different OSDs (iii) for the application objectives (i).

In the following case study, we apply NS–ERF to derive an airborne EC flux flight strategy that augments a network of EC flux towers for optimally addressing CHEESEHEAD19 science objectives: relating surface–atmosphere feedbacks over a 10 × 10 km study domain to energy balance closure and space–time scaling (Sect. 2.1). A preconceived network of 20 continuously operating EC flux towers forms CHEESEHEAD19's backbone for addressing these science objectives (Fig. 2). Tower placement within the study domain followed

## Numerical Simulation – Environmental Response Function OSSE

**Figure 3.** Visual glossary of the numerical simulation–environmental response function (NS–ERF) approach to observing system simulation experiments (OSSEs), consisting of three sets of elements: **(a)** definition elements, **(b)** realization elements, and **(c)** a benchmarking element. The text in Sect. 2.2 provides a detailed description of the interactions among individual NS–ERF elements.

a stratified random pattern, taking into account practical considerations including distance to road, suitable gaps in trees for a tower, and USFS-owned land. Individual towers were an average of 1.4 km from their nearest neighboring tower and an average of 3.5 km from the tall tower. The case study focuses on a strategy for airborne EC flux measurements because (i) they are central to linking the different scales of ground-based and spaceborne observations (Fig. 1), (ii) their flexibility provides an accessible mechanism to maximize joint information gain, and (iii) their flight safety and cost warrant careful planning. Notwithstanding, NS–ERF is broadly extensible beyond optimizing airborne EC flux measurements for large-scale field experiments, and at the end of this section we explore an adaptation to tower-EC-only natural climate solutions projects.

The application of NS–ERF to the CHEESEHEAD19 airborne design case study can be summarized as (i) generating virtual measurements, here in LES, (ii) joining information across disciplines and measurements in ERFs, and (iii) benchmarking candidate OSDs (Fig. 4). To obtain virtual measurements ahead of the actual field measurement campaign, we used LES to create a virtual atmosphere over the CHEESEHEAD19 domain for meteorological conditions characteristic of the measurement period (Fig. 4a). We then super-sampled this virtual atmosphere as it would be observed by 13 different yet simultaneous candidate OSDs over the duration of 2 h. Section 2.3 and 2.4 detail the specific implementation. ERFs then augment expensive and thus sparse response observations (e.g., fluxes from tower and airborne EC) with inexpensive, abundant biophysical driver observations (e.g., from meteorological stations and satellites; Fig. 4b). High-rate time–frequency decomposition and source area modeling facilitate mathematically rigorous data overlays among these response and driver observations at the minute and decameter scale. Machine learning then extracts a

driver-response process model from the resulting space- and time-aligned dataset. Ultimately, this driver-response process model complements the properties of response and driver observations in a response data product. In the present example, the response data products are decameter-scale sensible heat flux maps projected explicitly in space and time across the study domain. This is accomplished by executing the driver-response process model as a function of the driver inputs for each grid cell. Section 2.5 provides specific implementation details. Each candidate OSD resulted in a separate set of virtual observations that we independently processed through the ERFs. Finally, we benchmarked each candidate OSD by comparing the flux maps that ERF reconstructed from the virtual observations alone (Fig. 4c) to the original LES surface flux forcings (Fig. 4a). To judge the ability to reproduce the LES reference we used three optimality criteria (CR) directly related to the CHEESEHEAD19 science objectives, each ranging 0 %–100 %.

– CR1 is flux map spatial coverage, i.e., the percentage of grid cells across the study domain that ERF was able to reconstruct within the range of the virtual driver measurements (Sect. 2.5).

– CR2 is the energy balance ratio:

$$\text{EBR} = \frac{\langle F_{\text{H,ERF}} \rangle + \langle F_{\text{LE,ERF}} \rangle}{\langle F_{\text{H,LES}} \rangle + \langle F_{\text{LE,LES}} \rangle}, \tag{1}$$

with angle brackets indicating the horizontal average over all (reconstructed) grid cells over the study domain in the case of LES (ERF); $F_{\text{H}}$ and $F_{\text{LE}}$ indicate the sensible and latent heat flux, respectively. The numerator in Eq. (1) varies according to the different OSDs, and the denominator does not vary.

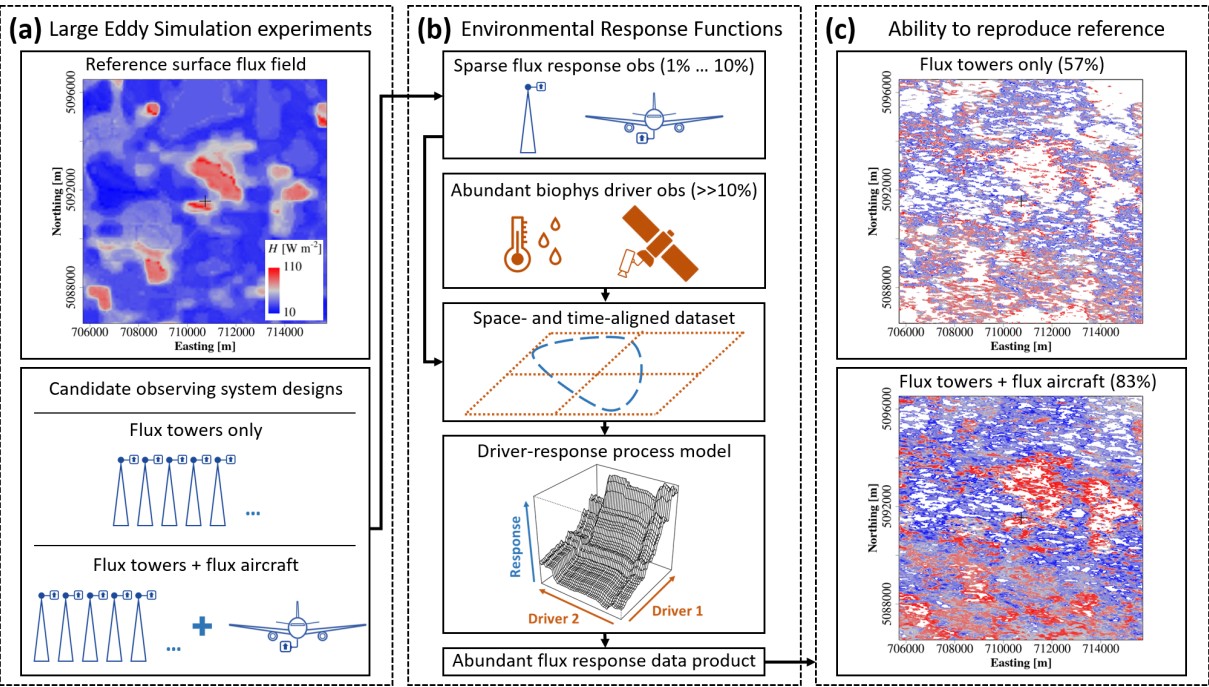

**Figure 4. (a)** To simulate different candidate OSDs ahead of the actual field measurement campaign, we used LES to obtain observations in virtual atmospheres over the CHEESEHEAD19 domain. **(b)** Environmental-response-function-related LES tower and airborne eddy covariance flux response observations at very high space–time resolution compared to LES ground-based, airborne, and spaceborne meteorological as well as surface driver observations. We used the extracted relationships to reconstruct fluxes explicitly across the study domain. **(c)** We then benchmarked the different OSD candidates against their ability to reproduce the LES reference in the form of flux grids that ERF reconstructed from the virtual observations alone. White grid cells denote areas outside the range of the virtual measurements, which let us directly estimate how well we sampled the domain for upscaling. Sections 2.2 and 2.5 provide additional details. Modified after Butterworth et al. (2021), copyright (2020), with permission from the American Meteorological Society to reuse under the CC BY 4.0 license (https://w3id.org/smetzger/Metzger-et-al_2021_OSSE-NS-ERF/cc-by-4.0, last access: 29 September 2021).

- CR3 is spatial patterning from pointwise Pearson correlation between the ERF-reconstructed flux maps and the corresponding LES forcings.

We then used the arithmetic mean and standard deviation to aggregate CR1–CR3 across flight patterns, flight sequences, and ultimately among themselves into a single score that directly corresponds to CHEESEHEAD19 science objectives (Sect. 3.3). It should be noted that the optimality criteria can be modularly adjusted for a range of applications. For example, NS–ERF OSSE could be applied to a natural climate solution project with tower-only (no airborne) EC measurements and the objectives to determine tower height, number, and location for economic implementation. In this case the single score could be a cost–benefit ratio consisting of optimality criteria from gridded carbon flux space–time coverage, carbon reduction potential relative to market price, and uncertainty. Proxies for photosynthetically active radiation, vertical gradients of $CO_2$, temperature and moisture, and relative measurement height in the ABL from satellites and station networks could provide the corresponding biophysical drivers (Xu et al., 2017). One should further distinguish the fact that the present OSSE study ex-

amines the potential to improve the information gain of the CHEESEHEAD19 airborne measurements through design choices. This is different from the scientific return of the actual CHEESEHEAD19 field campaign, which will become apparent in years to come as analysis and publication of the field data progress.

## 2.3 Design hypotheses and candidate observing system designs

We used NS–ERF to determine the sensitivity of the optimality criteria in Sect. 2.2 in response to two specific design hypotheses. These design hypotheses define the trade space between CHEESEHEAD19 science objectives, flight time constraints, and straightforward flight operation. With regard to flight patterns, we hypothesize that (H1) it is critical for airborne EC to measure perpendicular to the prevailing wind (e.g., Petty, 2021). And with regard to flight sequences, we hypothesize that (H2) within the flight time of a single research flight, it is more informative to fly a finely spaced pattern once compared to repeating a coarsely spaced pattern multiple times. To explore H1 and H2 we created candidate

OSDs in an LES (Sect. 2.4), consisting of a virtual EC flux tower network in combination with virtual airborne EC flight patterns and sequences.

The virtual EC tower network formed the backbone of each candidate OSD, and its horizontal distribution corresponded to the CHEESEHEAD19 stratified random grid pattern. A total of 19 virtual towers performed EC time series measurements at 49 m above ground, i.e., $26 \pm 13$ m higher compared to the real towers. The virtual AmeriFlux supersite tower at the center of the study domain measured at 49, 112, and 371 m above ground, i.e., $-6 \pm 17$ m lower compared to the real tower. These choices were a compromise among realism, bounding LES computational expense (10 m vertical resolution), and keeping the LES subgrid fluxes acceptably small ($< 1\%$) as suggested by Schröter et al. (2000), which will not be captured by the virtual EC flux computation. We analyzed 2 h of data for each of the 22 virtual tower-level combinations, or 44 h in total.

The virtual aircraft conducted EC space series measurements along grid flight patterns at 100 m above ground, identical to the measurement height proposed for the real aircraft. The grid consisted of 11 flight tracks each 25 km long, including six parallel flight tracks 2 km horizontally spaced from each other and five diagonal flight tracks in between (Fig. 2, blue arrows). To create the dataset for assessing H1 we formed the virtual flight patterns by letting multiple aircraft fly simultaneous grids along four different azimuth angles of the parallel tracks: east–west (E–W), north–south (N–S), southwest–northeast (SW–NE), and south–southwest–north–northeast (SSW–NNE). Here, the term *flight pattern* refers to a set of georeferenced waypoints. To assess H2 we further considered three permutations of the *flight sequence*, i.e. the order in which the waypoints of a given pattern are flown. (i) *Alternating* refers to flying a parallel track, then the downwind diagonal track, then the downwind parallel track, and so forth. (ii) *Outbound* refers to first completing all parallel tracks in downwind order, then flying back to the first parallel track and completing all diagonal tracks in downwind order. (iii) *Return* refers to first completing all parallel tracks in downwind order and then completing all diagonal tracks in upwind (return) order. To summarize, the virtual airborne EC dataset consisted of 3 flight sequences × 4 flight patterns × 11 flight tracks, or a total of 132 analyzed flight tracks spanning 3300 km of virtual airborne EC data.

Based on this super-sample we evaluated 13 candidate OSDs. Applying NS–ERF to 44 h of data from the virtual EC tower network alone provided the baseline OSD. Combining data from the virtual EC tower network with one of the 3 flight sequences × 4 flight patterns = 12 airborne EC combinations provided 12 alternative OSDs. Each of the alternative OSDs consisted of 44 h of site virtual tower EC data and 11 flight tracks × 25 km = 275 km virtual airborne EC data. This configuration allows us to evaluate the change in the optimality criteria (Sect. 2.2) for each of the 12 joint tower and aircraft alternative OSDs relative to the tower-only

baseline OSD. To summarize, the tower-only OSD yields a fixed baseline value for each of the spatial coverage, energy balance, and spatial patterning optimality criteria, and the alternative OSDs aim to maximally improve upon these baseline values by testing different flight strategies.

## 2.4 Large eddy simulations

We used the Parallelized LES Model (PALM) (Maronga et al., 2015, 2020) revision 4007 to simulate the atmosphere over the CHEESEHEAD19 domain. PALM solves the non-hydrostatic incompressible Boussinesq-approximated equations. We used the turbulent kinetic energy scheme of Deardorff (1980) for the subgrid model, a fifth-order scheme (Wicker and Skamarock, 2002) to discretize the advection terms, and a third-order Runge–Kutta scheme by Williamson (1980) for the time integration.

The aim of the simulation was to optimize OSDs for the upcoming field measurement campaign, meaning that the surface and atmospheric conditions were unknown. Hence, we simulated a single meteorological setting for a day with a well-developed summertime continental ABL on 12 August 2011, which is a typical situation for that region during the scheduled field measurement period. We considered the model surface to be flat, and as surface forcing we prescribed time-dependent, heterogeneous sensible and latent heat flux grids that Metzger et al. (2013b) have previously determined for this day. In an intermediary step we downscaled the original heat flux grids from 100 to 25 m horizontal grid spacing and from 1 h to LES time step, and we filled data gaps with the horizontally averaged flux. This approach provides a straightforward surface flux benchmark for evaluating NS–ERF results and forgoes the extensive data requirements of a coupled land surface model that would be difficult to fulfill prior to the actual field measurements. We then applied Monin–Obukhov similarity theory locally between the surface and the first vertical grid level as a surface boundary condition for the momentum equations. During the pre-field stage, information on forest size and patch distribution was insufficient to use a plant-canopy model for reliably describing momentum drag. Hence, we set a horizontally homogeneous roughness length of 2.0 m in the simulations because significant parts of the measurement site and its surroundings consist of forests. We then applied cyclic conditions at the lateral boundaries and provided initial vertical profiles of the horizontal wind components, potential temperature, and water vapor mixing ratio from nearby radiosonde observations (Green Bay Observations, station ID 72645; ~ 100 km to the southeast of the study domain). We assumed the observed westerly wind within the free atmosphere to be in geostrophic equilibrium and steady state, and we thus prescribed vertically constant profiles of the geostrophic wind components. For safety reasons the real-world flights were to take place on mostly cloud-free days, so clouds were not simulated.

With this setup, we simulated a $30 \times 30 \times 3$ km domain in the $x$, $y$, and $z$ direction, respectively, centered over the $10 \times 10$ km CHEESEHEAD19 domain. The grid spacing was 25 m in the horizontal direction and 10 m in the vertical direction. The simulation ran for 5 h (07:00–12:00 CST), the first 3 h of which were model spin-up time (07:00–10:00 CST). During the final 2 h (10:00–12:00 CST) we took virtual tower and aircraft measurements. At each virtual EC tower location, a virtual sensor sampled time series of potential temperature, mixing ratio, and vertical wind at the LES time step of 0.4 s. For each aircraft measurement, a similar virtual sensor moved along a predefined flight track at a ground speed of $82 \, \mathrm{m \, s^{-1}}$.

## 2.5 Environmental response functions

ERF employs time–frequency decomposition, source area modeling, and machine learning to join the information contained in multiscale environmental observations explicitly in space, time, and function (Metzger et al., 2013a). Compared to block averaging in traditional EC, spectral averaging in ERF facilitates orders-of-magnitude higher resolution of the resulting fluxes, here 1 min and 100 m vs. traditionally 30 min and $\sim 10$ km for tower and aircraft fluxes, respectively. This permits modeling the surface source area separately for each 1 min and 100 m flux response observation, thus further improving relatability to surface driver variability. For each 1 min and 100 m interval, the individual flux response observation is then stored alongside coinciding meteorological driver observations and source-area-averaged surface driver observations in a space- and time-aligned dataset. It is this high-resolution dataset that provides the necessary space–time matching and sample size to facilitate robust machine learning and subsequent flux map projection. Here we used ERF to reproduce the LES surface flux forcing from virtual EC towers, EC aircraft, and remote sensing observations (e.g., Xu et al., 2017; Serafimovich et al., 2018). These flux maps comply with observational assumptions that are not typically met from EC measurements alone, such as incorporation of mesoscale flows and spatial representativeness for the $10 \times 10$ km CHEESEHEAD19 target domain (Metzger, 2018; Xu et al., 2018, 2020).

ERF commenced with the high-rate time–frequency decomposed computation of the sensible and latent heat flux responses in the atmosphere. This step is based on the Morlet wavelet, while assuming constant ambient pressure in the LES. Spectral averaging over the wavelet cross-scalograms facilitated high temporal (tower: 1 min) and spatial (aircraft: 100 m) resolution of the resulting fluxes (Mauder et al., 2007a). To ensure that tower and aircraft fluxes are comparable in their inclusion of longwave mesoscale flows we applied a joint rectangular cutoff. The aircraft data limited the longwave transport scales, with the 25 km flight tracks equating to a maximum transport scale of $\sim 17$ km expressible by the wavelet cross-scalograms. We then applied Taylor's hypothesis (Taylor, 1915) with an average wind speed of 3–$5 \, \mathrm{m \, s^{-1}}$ to derive a corresponding tower cutoff scale of $\sim 1$ h. We time-matched the sensible and latent heat flux responses with the virtual observations of meteorological drivers consisting of potential temperature, water vapor dry mole fraction from mixing ratio, and relative measurement height in the ABL calculated from the potential temperature profile.

Next, we used source area modeling (Kljun et al., 2004; Metzger et al., 2012) to quantify the source area contributions to each 1 min tower and 100 m aircraft flux observation. The source area weights provided a linkage between the sensible and latent heat flux responses in the atmosphere and their spatially resolved drivers at the LES surface (available energy as a proxy for net radiation) as well as in the first vertical LES layer (near-surface temperature and moisture as a proxy for remote sensing observations). While near-surface temperature and moisture retain much of the horizontal spatial patterning, their amplitudes are reduced compared to actual surface temperature and moisture. This is exacerbated by the source area averaging applied here, and the combined effects on amplitude are evident, e.g., in Fig. 11. For simplicity, we used averages over the 2 h observation period for all spatially resolved drivers. The results are space- and time-aligned datasets consisting of the sensible and latent heat flux responses and their meteorological drivers in the LES atmosphere, as well as their spatially resolved drivers near the LES surface. The space–time-aligned dataset for the baseline OSD (tower-only) thus consisted of 22 virtual tower-level combinations with 2 h of data each at 1 min output resolution, resulting in 2640 observations. The space–time-aligned dataset for each of the 12 alternative OSDs (tower + aircraft) additionally consisted of 11 flight tracks with 25 km data each at 100 m output resolution, resulting in 2750 observations. It should be noted that this is the first application of ERF to combine flux response information across platforms, here flux tower and flux aircraft.

We then used boosted regression tree (BRT) machine learning to mine the information contained in the space–time-aligned datasets. The results were individual ERF process models for each OSD, separately for the sensible and latent heat flux responses as a function of their meteorological and surface drivers. Overall, we built the driver-response model structure in accordance with first principles: an energy source, from which fluxes result in accordance to Fick's law of (turbulent) diffusion along temperature and water vapor gradients, modulated with distance from the exchange surface. For example, we expressed the sensible heat flux response as a space–time function of the vertical temperature gradient. BRT then solved for the turbulent diffusion coefficient as a space–time function of available energy, modulated by vertical flux divergence and the vertical humidity gradient.

In the final step we projected the space–time explicit heat flux response maps to the median relative measurement height of the 49 m towers (0.16 of the ABL height). This is accomplished by providing the full complement of space–

time explicit drivers to the ERF process model. Specifically, we provided the spatially distributed near-surface temperature and moisture fields, the 2 h space–time median available energy across the $30 \times 30$ km domain, and the 2 h median meteorological drivers across all 20 virtual towers measuring at 49 m. This essentially equates to substituting the spatial information in the source areas with the distributed spatial information on near-surface temperature and moisture fields akin to remote sensing. While it would have been possible to resolve the meteorological drivers temporally and hence also the resulting heat flux maps, we used the 2 h aggregates to streamline the overall analysis. We also limited the ERF projection to interpolate but not extrapolate outputs, i.e. to only populate grid cells with driver combinations in the range of the virtual measurements. By doing so, the resulting data coverage lets us directly estimate how well we sampled the domain for upscaling. In total, we trained and projected 78 ERF process models consisting of two heat fluxes – sensible and latent heat – and 13 OSDs with three replicates each to constrain BRT tolerances.

## 3   Results

### 3.1   LES virtual experiments

As described in Sect. 2.4, the LES was forced using pre-existing surface sensible and latent heat fluxes across the domain. Figure 5 shows the prescribed surface sensible and latent heat fluxes at different points in time, which we used as lower boundary condition for the LES. The hourly input fluxes were interpolated in time to the LES time step. Surface heterogeneities with distinct patches in the surface sensible and latent heat flux are visible at multiple scales that vary in time and among the latent and sensible fluxes as well. Distinct surface heterogeneity patches are maintained over the entire simulation period, representing particular landscape patterns across the CHEESEHEAD19 domain.

Figure 6 shows the domain average initial and time-dependent vertical profiles of potential temperature, water vapor mixing ratio, and wind speed. These explain the virtual setup and provide an overview of the ABL structure: the model was initialized with the early morning profiles of potential temperature and mixing ratio, then left to develop its own equilibrium for this design case. The profile of potential temperature indicates a vertically well-mixed ABL, which heats up during the course of the day. Due to the strong capping inversion the ABL grows only slowly and reaches a height of about 400 m around noon, which is relatively low for a summertime convective ABL and further discussed in Sect. 4.1. The mixing ratio within the ABL also increases during the simulation due to the surface latent heat flux and due to entrainment of moist air from the above-lying free atmosphere, which exhibits higher values of mixing ratio than in the ABL. The profiles of the wind components indi-

cate northwesterly winds within the ABL during the morning hours, turning to westerlies later. Westerlies during the actual virtual measurement duration period from late morning until noon are required to investigate the candidate OSDs from Sect. 2.3.

Figure 7 shows a horizontal cross section of the instantaneous and time-averaged vertical wind component at a height of 100 m during the virtual measurement period at 11:00. Instantaneous updrafts and downdrafts ranging from $-2$ to $3 \, \mathrm{m \, s^{-1}}$ can be observed. The updrafts and downdrafts indicate elongated structures aligned with the mean wind direction at this height level. Even though the spatial organization of these structures is not strictly stationary in time due to the slightly changing wind direction (see Fig. 6), they can still be observed in the temporal average.

Figure 8 shows vertical profiles of the domain-averaged sensible and latent heat flux. Both flux profiles display a similar shape, linearly decreasing with height and reaching a minimum at the ABL top. These negative heat fluxes indicate entrainment of warm and moist air from the inversion into the ABL. This is supported also by Fig. 6, where the inversion layer exhibits a higher mixing ratio compared to the ABL. Figure 8 further shows that the subgrid-scale fluxes contribute less than 1 %–2 % to the total vertical transport at the measurement levels. This indicates that the vertical transport of heat and moisture is well resolved at these levels.

### 3.2   ERF retrievals

To create a space- and time-aligned dataset (Fig. 4b), ERF first calculates wavelet-decomposed EC fluxes directly from the high-frequency raw data. This facilitates inclusion of longer transporting scales compared to traditional EC, as well as unprecedented spatial and temporal resolution of the resulting fluxes (Fig. 9).

Next, ERF relates the time- and space-resolved EC flux responses in the atmosphere to biophysical drivers at the surface (Fig. 10), such as near-surface temperature and near-surface moisture. In the present application near-surface temperature and near-surface moisture are taken from cross sections at the vertical LES level closest to the surface. In real-world ERF applications, these are substituted with space-borne and airborne remote sensing data products or reanalysis data (e.g., Serafimovich et al., 2018). This facilitates mathematically rigorous data overlays among response and driver observations at the minute and decameter scale. The result is a space–time-aligned dataset for each virtual EC tower and for each virtual EC flight track. Both the tower and airborne EC datasets comprise the same variables in identical units and were processed to ensure cross-platform compatibility and avoid biases (Sect. 2.5). This allows combining the virtual tower EC results and corresponding virtual airborne EC results into a single space–time-aligned dataset for each of the 12 alternative TS2 OSDs.

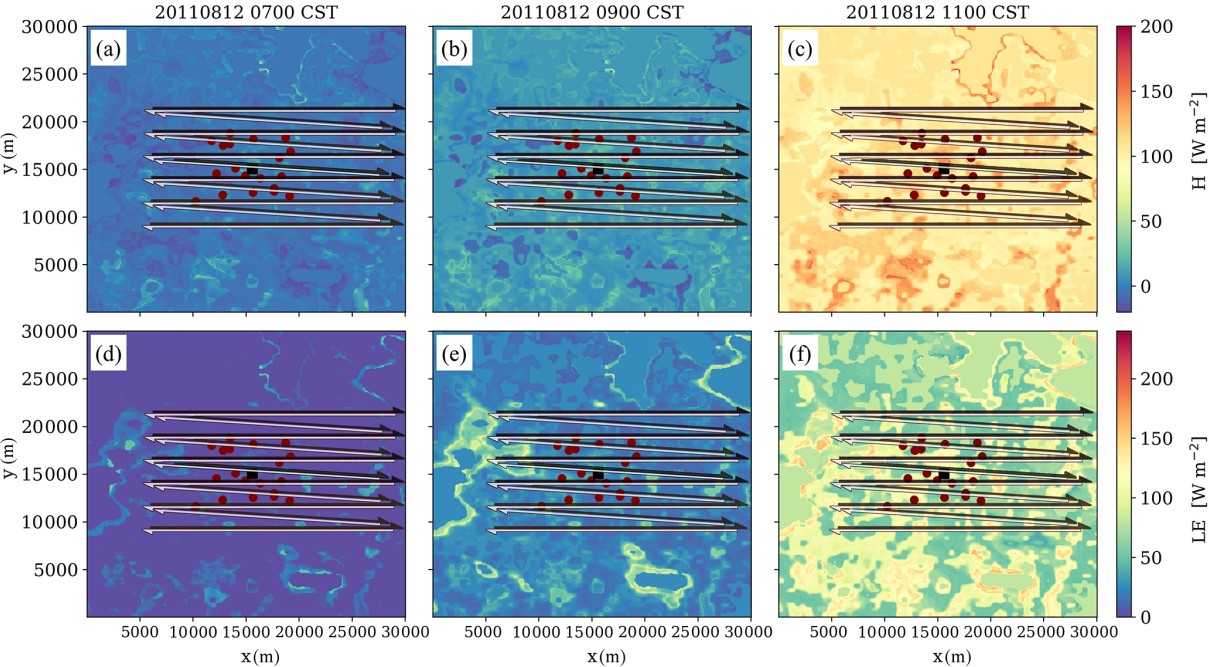

**Figure 5.** Time sequence of **(a–c)** the spatially heterogeneous surface sensible heat flux and **(d–f)** latent heat flux prescribed as LES lower boundary conditions. Superimposed red dots indicate candidate EC tower locations, alongside UWKA candidate flight tracks (west–east pattern).

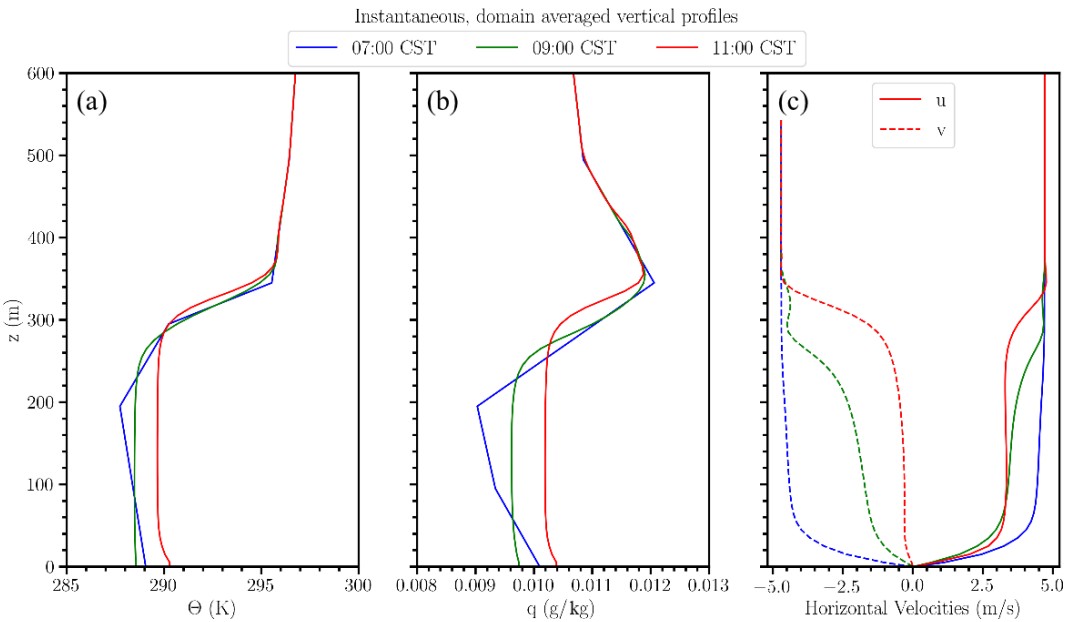

**Figure 6.** LES domain-averaged vertical profiles of **(a)** potential temperature, **(b)** water vapor mixing ratio, and **(c)** horizontal wind velocities at different simulation times.

The ERF machine learning component then extracts a driver-response process model from each of the combined space- and time-aligned datasets. Here, we trained a total of 78 machine learning models. These consisted of 13 candidate OSDs × 2 fluxes (sensible and latent heat) × 3 repli-cates (to quantify stochastic uncertainty in the response data products). Figure 11 shows example driver-response surfaces for sensible and latent heat flux as a function of their prin-cipal drivers, energy input, near-surface temperature, and near-surface moisture. This exemplifies in reduced dimen-

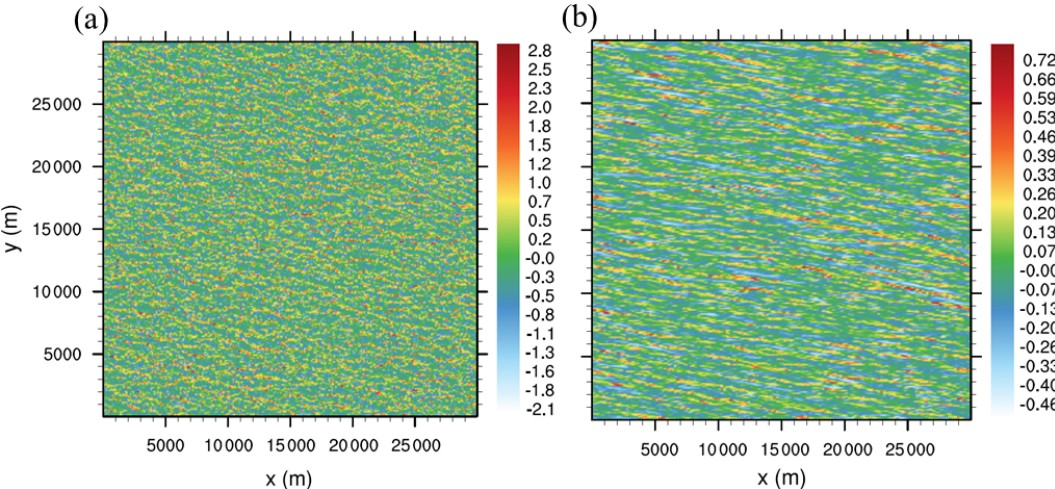

**Figure 7.** LES $x$–$y$ cross section of **(a)** instantaneous and **(b)** 30 min time-averaged vertical velocity at a height of 100 m at 11:00 CST simulation time.

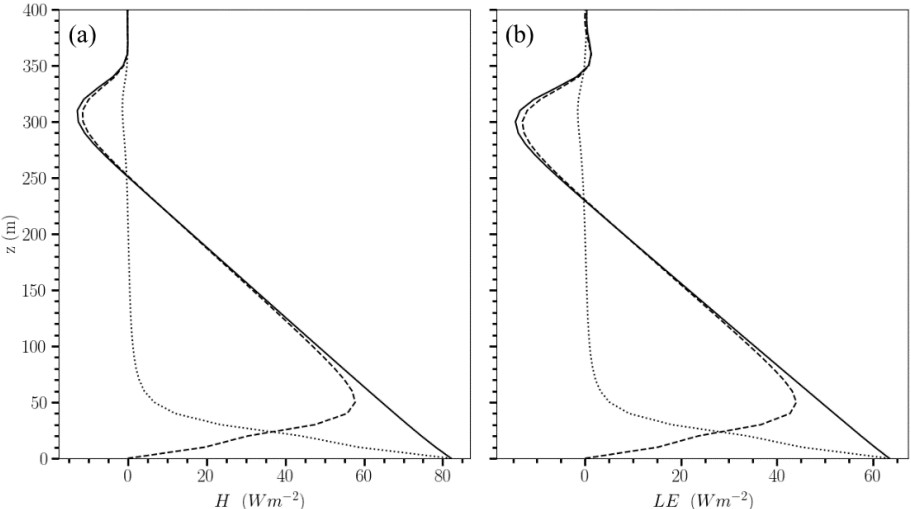

**Figure 8.** Domain-averaged vertical flux profiles of **(a)** sensible heat and **(b)** latent heat at 11:00 CST simulation time. The solid lines show the total simulated fluxes, consisting of resolved fluxes (dashed lines) and subgrid parameterized fluxes (dotted lines).

sionality how the turbulent diffusion coefficient connects the drivers to the flux response within the physics-guided ERF model structure. In Fig. 11a the sensible heat flux increases primarily with near-surface temperature and secondarily with energy input. The relationship reaches a plateau around 290.3 K, which deviates from a one-dimensional, monotonic, and linear gradient–flux relationship, indicative of additional feedback processes. Conversely, in Fig. 11b the latent heat flux increases primarily with energy input and secondarily with near-surface moisture, with monotonic and approximately linear relationships across the range of drivers.

Ultimately, the physics-guided ERF driver-response process model complements the properties of response and driver observations in a response data product. In the present example the response data products are decameter-resolution

sensible heat flux maps projected explicitly in space and time across the study domain (Fig. 12). We projected the flux maps for the tower-only space–time-aligned dataset (baseline OSD; Fig. 12a) and then separately for each of the 12 joint tower and aircraft space–time-aligned datasets (alternative OSDs; Fig. 12c). Now the flux maps that ERF reconstructed from the virtual measurements alone can be compared to the original LES surface flux forcings (Fig. 12b).

### 3.3 Evaluation of design hypotheses

The ERF-derived flux maps alongside the LES surface forcing in Fig. 12 allow us to assess the design hypotheses (Sect. 2.3) as a function of the different candidate OSDs. For this purpose, we evaluated the change in the optimality cri-

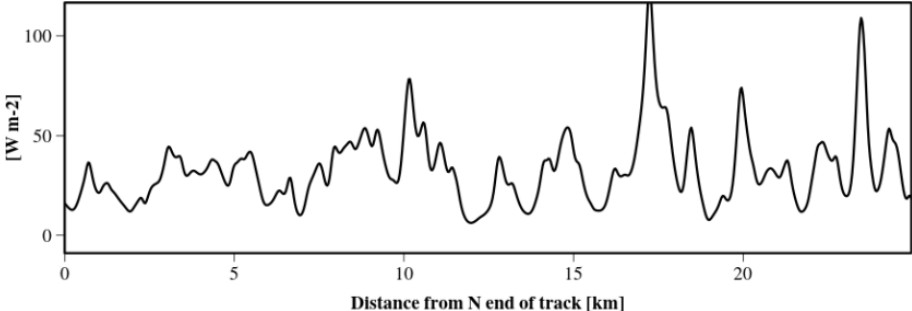

**Figure 9.** Space-resolved sensible heat flux from high-rate space-scale decomposition of virtual airborne measurements.

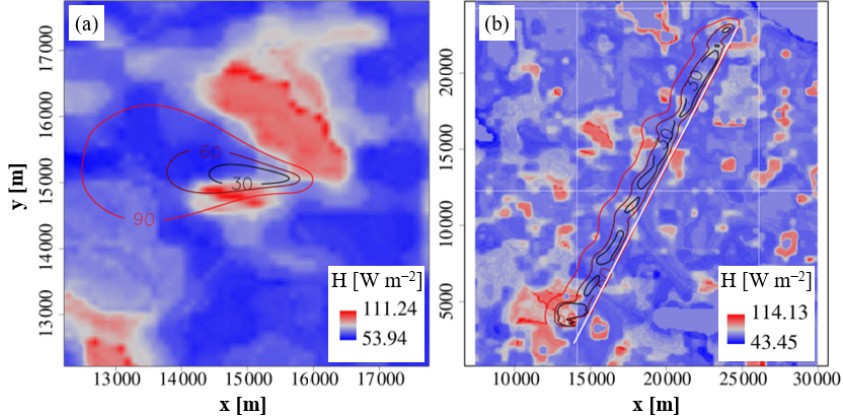

**Figure 10.** Source area modeling (30 %, 60 %, 90 % contour lines) relates observations across platforms and representations by determining the surface sources of the time- and space-resolved EC fluxes, here superimposed over the LES sensible heat flux surface forcing. **(a)** Virtual AmeriFlux supersite tower at the center of the study domain at a measurement height of 112 m. **(b)** Virtual UWKA flight track at a measurement height of 100 m.

teria (Sect. 2.2) for each of the 12 joint tower and aircraft alternative OSDs relative to the tower-only baseline OSD. In the following (Tables 1–3) we performed all aggregations using arithmetic mean and standard deviation operators. In all cases the aggregations include two fluxes (sensible and latent heat) with three machine learning replicates each, plus additional aggregation as specified.

In response to the first design hypothesis, H1, we address the question of how critical it is for airborne EC to measure perpendicular to the prevailing wind. Table 1 shows the results for each optimality criterion as a function of the aircraft track angle on the mean wind direction, aggregated over all three flight sequences. We can see that track angles in the range of $90 \pm 45°$ on the mean wind direction yield limited improvement in spatial coverage ($23.3 \% \pm 1.8 \% – 25.6 \% \pm 0.1 \%$) compared to wind-parallel patterns ($0°$; $20.9 \% \pm 1.9 \%$). However, within the same range of track angles the improvement in energy balance ratio and spatial patterning approximately double to octuple (Table 1, italic font).

The improvement in spatial patterning when adding wind-parallel flights to the tower network is limited to $13.7 \% \pm 9.2 \%$ compared to $18.3 \% \pm 15.2 \% – 34.6 \% \pm 3.3 \%$ for adding flights with a $45–90°$ aircraft track angle on the mean wind. Irrespective of the track angle, the observations along a flight track are never entirely independent from each other due to along- and cross-wind dispersion. For wind-parallel flights, Fig. 13a indicates a high degree of source area overlap and thus self-correlation among the observations, resulting from strong along-wind dispersion along the flight track. In contrast, Fig. 13b shows fewer overlapping source areas along the flight track of wind-perpendicular flights, with the comparatively weaker cross-wind dispersion now controlling the overlap. The latter strategy results in observations that capture more independent samples and spatial variability. Thus, the dominating mode of atmospheric dispersion with respect to the aircraft track angle helps to explain the differences in the spatial patterning optimality criterion. For our study setup we reach a critical overlap resulting from the combined effects of along- and cross-wind dispersion at track angles shallower than $90 \pm 45°$ on the mean wind direction.

Furthermore, at the virtual aircraft flight height of 100 m the time-averaged vertical wind cross section in Fig. 7b

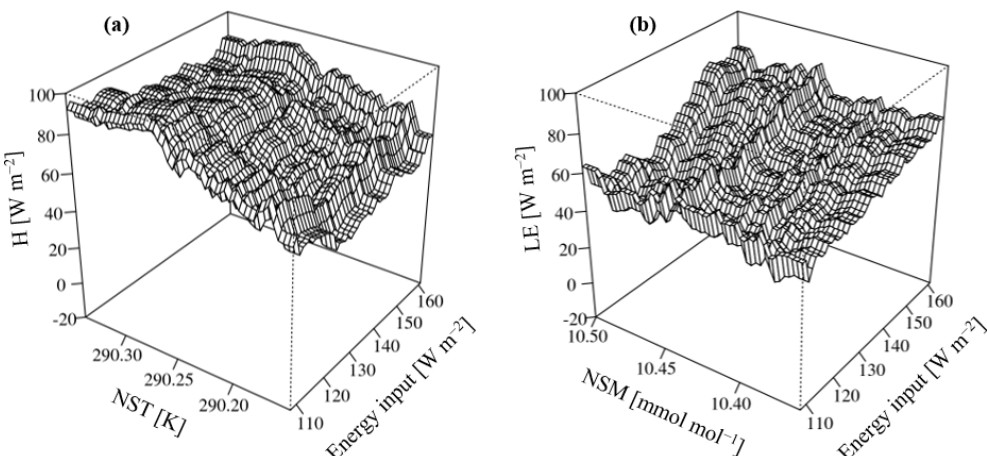

**Figure 11.** Example ERF surfaces. **(a)** Sensible heat flux as a function of source-area-averaged energy input and near-surface temperature (NST from the first vertical LES layer; Sect. 2.5). **(b)** Latent heat flux as a function of source-area-averaged energy input and near-surface moisture (NSM from the first vertical LES layer; Sect. 2.5). For this visualization, all other drivers are kept at their median value.

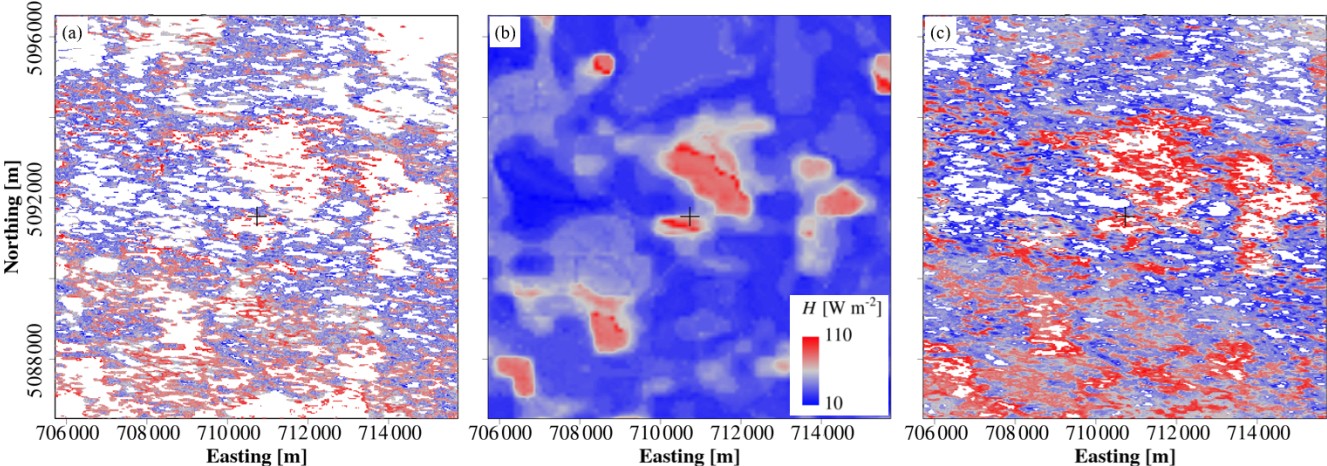

**Figure 12.** Example ERF data products: sensible heat flux maps independently derived for **(a)** the tower-only space–time-aligned dataset and **(c)** for the joint tower and aircraft space–time-aligned dataset, alongside **(b)** the LES reference surface flux field.

shows organized structures that are elongated in the mean wind direction. These organized structures are among the most promising leads to explain the frequently observed non-closure of the energy balance, in particular from tower EC measurements (Mauder et al., 2020). These structures consist of more spatially expansive though weaker subsidence zones and more spatially limited though stronger convection zones (Lenschow and Stankov, 1986; Moeng and Rotunno, 1990; Etling and Brown, 1993; Kanda et al., 2004; Petty, 2021). So, when applied to aircraft EC measurements, wind-parallel flights are more likely to occur along the subsidence zones than along the convection zones. This helps explain why adding wind-parallel flights to the tower network yields only a limited improvement of the energy balance ratio criterion (1.7 % ± 1.4 %). Conversely, wind-perpendicular flights trend toward observing combinations of subsidence

and convection zones that approximately balance the atmospheric conservation of mass and energy. This explains the eightfold improvement (12.8 % ± 3.1 %) when adding wind-perpendicular flights to the tower network compared to wind-parallel flights. Between these two extreme cases, adding the flights with 45 and 60° track angles to the tower network still yields an approximately fourfold improvement (6.4 % ± 4.7 %–6.4 % ± 5.3 %) over the wind-parallel flights.

Next, we address the design hypothesis, H2, i.e., whether it is more informative to fly a finely spaced pattern once or to fly a coarsely spaced pattern multiple times. Table 2 shows that the spatial coverage and energy balance ratio criteria are not particularly sensitive to the flight sequence. One exception is the particularly high and consistent improvement in the spatial patterning performance criterion of the alternating sequence (29.1 % ± 5.4 %; Table 2, italic font). It is the

**Table 1.** Percent improvement of the joint tower and aircraft EC alternative OSDs relative to the tower-EC-only baseline OSD, aggregated over all three flight sequences. The results are shown as a function of the optimality criterion (rows) and aircraft flying the grid pattern in various track angles on the mean wind direction (columns). Bold font highlights marked improvements that are further discussed in the text.

| Optimality criterion | All angles | 0° | 45° | 60° | 90° |
|---|---|---|---|---|---|
| Spatial coverage | 23.6 % ± 2.2 % | 20.9 % ± 1.9 % | 24.7 % ± 0.8 % | 23.3 % ± 1.8 % | 25.6 % ± 0.1 % |
| **Energy balance ratio** | 6.8 % ± 5.3 % | 1.7 % ± 1.4 % | **6.4 % ± 5.3 %** | **6.4 % ± 4.7 %** | **12.8 % ± 3.1 %** |
| **Spatial patterning** | 23.2 % ± 11.7 % | 13.7 % ± 9.2 % | **34.6 % ± 3.3 %** | **26.2 % ± 6.8 %** | 18.3 % ± 15.2 % |

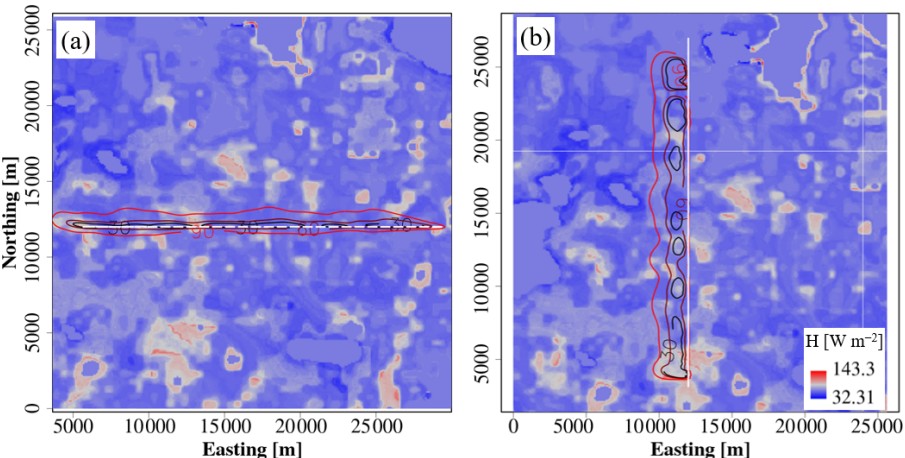

**Figure 13.** Example virtual flight tracks and their 30 %, 60 %, and 90 % source area contours superimposed over the LES sensible heat flux surface forcing (W m$^{-2}$). **(a)** Wind-parallel flights sample source areas that are elongated along the flight track, leading to considerable overlap. **(b)** Wind-perpendicular flights sample less overlapping source areas along the flight track and thus capture more independent samples and spatial variability.

only sequence that "carpets" the CHEESEHEAD19 domain wall to wall at fine time and space increments. All other sequences progress in coarser increments, such as first completing all parallel flight tracks and then revisiting the interspersed diagonal flight tracks. In the context of the 2 km horizontally spaced parallel flight tracks, Xu et al. (2017) report a 411 ± 88 m characteristic surface length scale of landscape elements in the CHEESEHEAD19 domain. The finer increments of the parallel–diagonal alternating sequence let ERF relate drivers and responses closer to the characteristic surface length scale and thus to reproduce the spatial patterning.

To summarize, flight patterns with a track angle in the range of 90 ± 45° on the mean wind direction yielded approximately double the performance improvement of wind-parallel patterns. This finding is irrespective of the flight sequence but most consistent for the alternating flight sequence (21.6 % ± 11.5 %–22.6 % ± 9.4 %; Table 3, italic font). Compared to the worst-case combination of wind-parallel flight patterns with the outbound flight sequence (9.6 % ± 11.1 % improvement) this equates to doubling the information gain.

### 3.4 Field campaign resources

Flying the grid pattern in the alternating sequence provided the best and most consistent results, while also being the most straightforward sequence for operational implementation. Further, to satisfy the 90 ± 45° track angle on the mean wind condition we derived three rotationally symmetric sets of waypoints at 60° increments (Fig. 14). Flying the numbered waypoint in ascending order results in three alternating flight sequences: SE1, SW1 TS3, and WE1 with 330°, 30°, and 90° azimuth of the parallel tracks, respectively. Owing to rotational symmetry, flying the numbered waypoints in descending order results in three additional alternating flight sequences: SE2, SW2 TS4, and WE2 with 150°, 210°, and 270° azimuth of the parallel tracks, respectively. Reversing the waypoint order allows the aircraft to progress through the flight tracks in downwind order for any given mean wind direction. This aims to reduce the space–time ambiguity resulting from airborne EC observing different surface conditions over hundreds of kilometers, while at the same time the diurnal cycle progresses over the course of several hours. Lastly, to avoid the town and airfield of Park Falls, WI, immediately west of the CHEESEHEAD19 domain, we shifted the WE1/WE2 set of waypoints 5 km to the east (Fig. 14c).

To support daily flight planning we distilled the six alternating flight sequences into a flight planning wind rose (Fig. 15). There we implemented the track angle condition by superimposing over a wind rose the wind sector aligned

https://doi.org/10.5194/amt-14-1-2021 Atmos. Meas. Tech., 14, 1–26, 2021

**Table 2.** Percent improvement of the joint tower and aircraft EC alternative OSDs relative to the tower-EC-only baseline OSD, aggregated over all four aircraft track angles on the mean wind direction. The results are shown as a function of the optimality criterion (rows) and aircraft flying the grid pattern in various sequences (columns). The bold font highlights a marked improvement that is further discussed in the text.

| Optimality criterion | All sequences | **Alternating** | Outbound | Return |
|---|---|---|---|---|
| Spatial coverage | 23.6 % ± 2.2 % | 23.0 % ± 3.0 % | 23.5 % ± 1.8 % | 24.4 % ± 1.9 % |
| Energy balance ratio | 6.8 % ± 5.3 % | 7.9 % ± 5.3 % | 6.0 % ± 6.6 % | 6.6 % ± 5.4 % |
| **Spatial patterning** | 23.2 % ± 11.7 % | **29.1 % ± 5.4 %** | 14.9 % ± 15.7 % | 25.6 % ± 9.0 % |

**Table 3.** Percent improvement of the joint tower and aircraft EC alternative OSDs relative to the tower-EC-only baseline OSD, aggregated into a single score over all optimality criteria. The results are shown as a function of aircraft flying the grid pattern in various sequences (rows) and track angles on the mean wind direction (columns). Bold font highlights marked improvements that are further discussed in the text.

| Flight sequence | All angles | 0° | **45°** | **60°** | **90°** |
|---|---|---|---|---|---|
| All sequences | 17.9 % ± 10.8 % | 11.6 % ± 8.8 % | 19.9 % ± 12.2 % | 16.9 % ± 10.0 % | 18.5 % ± 9.6 % |
| **Alternating** | 20.0 % ± 10.2 % | 13.3 % ± 11.4 % | **22.6 % ± 9.4 %** | **21.6 % ± 11.5 %** | **22.4 % ± 11.7 %** |
| Outbound | 14.8 % ± 11.6 % | 9.6 % ± 11.1 % | 21.3 % ± 15.9 % | 13.8 % ± 11.2 % | 14.5 % ± 11.9 % |
| Return | 18.9 % ± 10.7 % | 13.4 % ± 10.3 % | 21.8 % ± 17.7 % | 20.5 % ± 10.2 % | 19.8 % ± 6.2 % |

$90 \pm 45°$ to the parallel tracks of each of the six alternating flight sequences. This allows determining the appropriate flight sequence as a function of the forecasted mean wind direction. For example, if experiencing southerlies (180°) the most suitable flight sequence is WE2. Owing to rotational symmetry, the wind sector for each flight sequence overlaps with each of its two neighbors by 30°. This provides a margin for accommodating changing synoptic conditions. For example, if experiencing south-southwesterlies (210°) in the morning the WE2 and SE1 [TS5] flight sequences would be equally suitable. If, however, the mean direction is forecasted to shift to westerlies (270°) in the course of the day the SE1 [TS6] flight sequence simplifies flight operation by satisfying the $90 \pm 45°$ track angle on the mean wind condition with a single flight sequence for a given day.

## 4 Discussion

Upon deriving the NS–ERF framework, we identified an optimal OSD for the CHEESEHEAD19 case study that promises to more than double information gain. Here we initially discuss how these numerical gains relate to improving our potential for addressing CHEESEHEAD19 science objectives and their limitations. We then examine how the resulting field campaign resources improved flight operation and crew safety by an order of magnitude. Lastly, we reflect on our findings in light of existing design approaches, provide general recommendations for future OSDs, and discuss remaining challenges and future work.

### 4.1 Optimizing the CHEESEHEAD19 observing system design

NS–ERF used three optimality criteria (Sect. 2.2; CR1 spatial coverage, CR2 energy balance ratio, CR3 spatial patterning) that we tailored to represent CHEESEHEAD19's science objectives numerically. Furthermore, we identified two specific design hypotheses that we postulate the science objectives, and hence optimality criteria, to be sensitive to (Sect. 2.3; H1 track angle on the mean wind, H2 fine vs. coarse flight sequence). CHEESEHEAD19's first science objective, O1, is to show that higher surface heterogeneity promotes energy transport in atmospheric mesoscale eddies. Our potential to address this science objective increases with the truthful reproduction of CR1 surface flux spatial coverage and CR3 spatial patterning. NS–ERF allowed us to assess changes in these criteria resulting from the different OSDs by comparing ERF flux map reproductions to the original LES surface flux forcing. We found that CR1 spatial coverage is largely insensitive to H1 track angle on the mean wind (Table 1) and H2 fine vs. coarse flight sequence (Table 2). Conversely, CR3 spatial patterning proved to be highly sensitive to H1 track angle on the mean wind. Track angles in the range of $90 \pm 45°$ on the mean wind yielded double to triple improvements over wind-parallel flights (Table 1). Similarly, we showed that CR3 spatial patterning is sensitive to H2 fine vs. coarse flight sequence (Table 2). The finely spaced "alternating" sequence yielded the highest and most consistent spatial patterning improvements of about 50 % over the other flight sequences.

CHEESEHEAD19's second science objective, O2, aims to account for energy transport in mesoscale eddies and deter-

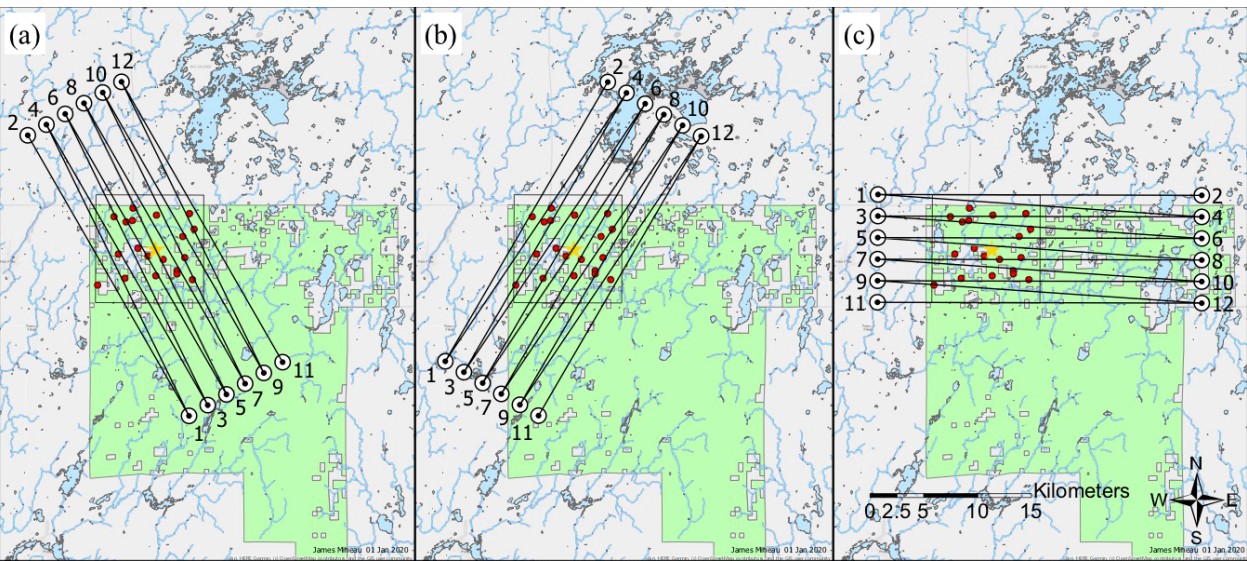

**Figure 14.** Three sets of waypoints define three distinct flight patterns, named after the relative location of their first two waypoints: **(a)** south-west (SW), **(b)** southeast CE3 (SE), and **(c)** west–east (WE). Flying the numbered waypoints either in ascending order (SW1, SE1, WE1) or descending order (SW2, SE2, WE2) resulted in six distinct flight sequences that maximize data coverage under different wind conditions. Map credit: James Mineau, University of Wisconsin–Madison.

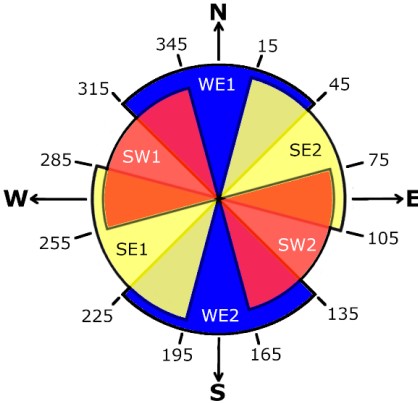

**Figure 15.** Flight planning wind rose to choose the appropriate flight sequence as a function of the forecasted mean wind direction. Owing to rotational symmetry, the wind sector for each flight sequence overlaps each of its two neighbors by 30°. This provides a margin for accommodating changing synoptic conditions. TS7

mine the "true" surface energy balance to improve model representation of subgrid processes. As such, our potential to address this science objective hinges on improved closure of the CR2 energy balance ratio, which proved to be highly sensitive to the H1 track angle on the mean wind. Here, track angles in the range of $90 \pm 45°$ on the mean wind yielded quadruple to octuple improvements over wind-parallel flights (Table 1). On the other hand, the CR2 energy balance ratio was comparatively insensitive to H2 fine vs. coarse flight sequence (Table 2).

Finally, CHEESEHEAD19's third science objective, O3, aims to demonstrate that ERF yields representative fluxes at model grid scale regardless of mesoscale eddies. ERF's potential to reproduce the surface flux is thus directly related to the combination of all criteria discussed above: CR1 spatial coverage, CR2 energy balance ratio, and CR3 spatial patterning. From aggregating over all optimality criteria into a single score, we demonstrated that overall improvement is highly sensitive to H1 track angle on the mean wind, as also shown by Petty (2021). Flight patterns with a track angle in the range of $90 \pm 45°$ on the mean wind yielded approximately double the performance improvement of wind-parallel patterns (Table 3). This finding is less sensitive to H2 fine vs. coarse flight sequence, though it is most consistent for the finely spaced alternating sequence. Overall, this combination doubles the information gain compared to the worst-case combination of wind-parallel flight patterns with the outbound flight sequence. Consequently, the study hypothesis that the CHEESEHEAD19 information gain is sensitive to NS–ERF optimization can be accepted. On the other hand, the design hypothesis H1 that it is critical for airborne EC to measure perpendicular to the prevailing wind should be rejected, as up to $\pm 45°$ tolerance yielded comparable results for the CHEESEHEAD19 science objectives. Lastly, the design hypothesis H2 that it is more informative to fly a finely spaced pattern should be accepted, with most consistent results for the finely spaced alternating sequence.

The field measurement campaign resources (Sect. 3.4) encapsulate these findings into only three sets of waypoints and six alternating flight sequences incremented at 60° azimuth. These provide a balance between scientific fidelity

and flight crew safety. On the one hand, the small number of waypoints and flight sequences is sufficient to maximize the CHEESEHEAD19 information gain by enabling adherence to the $90 \pm 45°$ track angle on the mean wind condition at all times. Furthermore, the 60° incrementation in combination with the $\pm 45°$ tolerance on perpendicularity to the mean wind provides 30° overlap among flight sequences to support decision-making during nonstationary synoptic conditions. On the other hand, the parsimonious number of only six flight sequences and the even smaller number of three sets of waypoints simplify flight planning and navigation. In combination with entirely avoiding the town and airfield of Park Falls this promotes flight crew safety by an order of magnitude compared to the originally envisioned 48 flight sequences. Specifically, it frees up the flight crew from arduous navigation details, thus reducing fatigue, increasing awareness during the 100 m low-level flight maneuvers, and ultimately reducing the margin for human error. Due to its vicinity to the 400 m tall tower and related in-flight safety concerns the central diagonal flight track in this study was not performed during the actual CHEESEHEAD19 field measurement campaign.

After deriving the above strategy, we detected an inconsistency in the vertical humidity profile that we used to initialize the LES and thus to produce the NS–ERF virtual observations. Specifically, we had erroneously added a positive vertical humidity gradient at 350 m above ground instead of the negative vertical humidity gradient typically observed by radiosondes adjacent to the CHEESEHEAD19 domain (sign reversal). As a result, we detected an ABL height of only 500 m in the LES, while field observations typically indicate > 1 km during similar conditions around the CHEESEHEAD19 domain. In addition, during 11:00 CST we detected a small entrainment flux of humid air from above the LES ABL into the drier air below, which is not typical of summertime convective ABL conditions around the CHEESEHEAD19 domain. In the subsequent stages of NS–ERF, we used virtual tower EC observations at 49 m ($N = 20$ towers), 112 m ($N = 1$ tower), and 371 m ($N = 1$ tower) above ground, as well as virtual airborne EC observations at 100 m above ground. At any given time, the surface fluxes prescribed in the LES were orders of magnitude larger compared to the entrainment fluxes. Hence, the surface fluxes dominated all virtual tower and airborne observations, possibly with the exception of the 371 m tower that, however, still reported an average upward latent heat flux of $2.2 \pm 6.6 \, \mathrm{W\,m^{-2}}$. While the uncharacteristically shallow ABL height results in 2 to 3 times more pronounced vertical flux divergence, NS–ERF accounts for this by utilizing the relative measurement height in the ABL as an ERF driver. Furthermore, the study design cancels possible residual impacts on the CR2 energy balance ratio by normalizing all results for the alternative OSDs (tower and aircraft) with the results for the baseline OSD (tower only). To summarize, the erroneous vertical humidity gradient resulted in a modeled LES atmosphere that was less

specific to the CHEESEHEAD19 domain than originally envisioned. However, this should have little to no bearing on the general findings that informed the CHEESEHEAD19 OSD owing to ERF accounting for vertical flux divergence and the normalized study design. If at all, surface heterogeneity scales across the CHEESEHEAD19 domain are more realistically reproduced compared to the idealized LES runs in many previous studies (e.g., Kanda et al., 2004; Sühring et al., 2018; Xu et al., 2020).

Strictly speaking, the CHEESEHEAD19 case study is limited to 2 h of LES data for a single meteorological setting and omission of clouds. While clouds and other variations will certainly change the entire turbulent structure of the ABL, this study also constituted a race against the clock to provide numerical insights in time to support the CHEESEHEAD19 field experiment design. In front of that background, we chose to focus on an LES run that is both typical for the region and/or season and one that likely generates significant heterogeneity without the added expense of dealing with clouds. Further, considering that for safety reasons the real-world flights were to take place on mostly cloud-free days, we believe the selected case provides a useful case study for how much airborne flight track choices influence our ability to address the CHEESEHEAD19 science objectives. With more lead time and computational resources it is possible to realize additional LES runs.

Overall, the application of the NS–ERF-derived field measurement resources enabled the successful acquisition of 14 400 km airborne data by the UWKA aircraft (Paleri et al., 2019). The 24 UWKA research flights and their on-site planning covered 480 flight tracks during 72 h of flight time and three CHEESEHEAD19 intensive observation periods (Butterworth et al., 2021). This further demonstrates the successful application of NS–ERF and its utility for determining concise and adaptive field measurement resources that optimize the effectiveness and safety of research flights. With the potential to improve the information gain of CHEESEHEAD19 airborne measurements clearly evident from this 2 h case study, it will be instructional to witness the true scientific return as analyses of the actual CHEESEHEAD field campaign measurements commence.

## 4.2 Benefits for coordinated environmental observations

NS–ERF extends previous approaches to designing large-scale field campaigns such as FIFE, BOREAS, NOPEX, LITFASS-98, LITFASS-2003, MAGIM, and ScaleX (Sect. 1). Specifically, NS–ERF not only utilizes but also fully contextualizes expert knowledge by conducting virtual pre-field measurements in NSs and using evidence-driven ERF to quantify the resulting information contents.

For decades, NS "data from knowledge" studies have investigated surface–atmosphere interactions including energy balance processes (Deardorff, 1972; Wyngaard and Brost,

1984; Etling and Brown, 1993; Kanda et al., 2004; Sühring and Raasch, 2013). Indeed, NSs have become useful to contextualize observational phenomena with increasingly complex feedbacks, including natural resources such as air quality (Khan et al., 2021; Zhang et al., 2021). However, the resulting data are detailed to a point at which it becomes challenging to fully utilize the provided information for extracting and describing the phenomena of interest. Xu et al. (2020) point to a possible solution to this dilemma by complementing detailed LES outputs with the dedicated ERF "knowledge from data" approach. Here, we took a next step and demonstrated the usefulness of the NS–ERF symbiosis for designing real-world field measurements. Using NSs for OSD has been a rare application to date (Cortina and Calaf, 2017; Gehrke et al., 2019), and to our knowledge the present study is the first of its kind that empowers investigators to harness the combined power of complementing NSs and data mining for this purpose.

OSSEs are widely used in the Earth system sciences to predict the performance of major, long-term research equipment and facility investments (Hargrove and Hoffman, 2004; Masutani et al., 2010; Zhang and Pu, 2010; Lucas et al., 2015; Hoffman and Atlas, 2016; Park and Kim, 2020). The NS–ERF symbiosis now provides the necessary resolution of time, space, and processes to make the power of OSSEs also accessible for designing field measurements at smaller and previously inaccessible scales. Specifically, the CHEESE-HEAD19 case study at the interface of mesoscale and microscale meteorology demonstrated a new degree of realism and explicitness in maximizing the joint information from ground-based, airborne, and spaceborne observations for scaling and modeling.

Building on this central property, NS–ERF is modularly extensible in multiple directions. For example, NS–ERF can integrate new types of observations in addition to tower, aircraft, and satellite observations, so long as their source areas are readily quantifiable. Examples are remote sensing of the atmosphere (Wulfmeyer et al., 2018; Helbig et al., 2020) and soil and biometric observations (Metzger et al., 2019b). This provides a promising avenue to maximize cross-disciplinary, cross-project, and ultimately cross-institutional synergies, such as through simulating the design of supersites that envision synergizing diverse observational infrastructures including from the US National Science Foundation's National Center for Atmospheric Research and National Ecological Observatory Network (Metzger et al., 2019a). Then upon completion of the planned field measurements, the real-world data can immediately substitute the NS "data from knowledge" module while the ERF "knowledge from data" module continues to perform the intended end-to-end analyses. NS–ERF thus provides a framework to prepare and test the quantification of science objectives well ahead of the actual field measurements, thus reducing latency from field data capture to knowledge creation. More generally, NS–ERF can extend to any sort of study in which spatially and/or temporally

**Table 4.** Labor and computing resources utilized for deriving the CHEESEHEAD19 observing system design, separately for large eddy simulations (LESs) and environmental response functions (ERFs) `CE4`.

| Resource | LES | ERF |
|---|---|---|
| Total labor | 180 h | 300 h |
|     Conceptualization | 30 h | 70 h |
|     Setup | 110 h | 130 h |
|     Analysis | 20 h | 50 h |
|     Interpretation | 20 h | 50 h |
| Computing architecture | High-performance | High-throughput |
|     CPU hours | 230 000 | 7000 |
|     CPUs | 1800 | 2–16 |
|     Memory | 1.8 TB | 16–128 GB |
|     Data produced | 210 GB | 4 GB |

sparse observations of a surface or atmospheric property X need to be combined with spatially and/or temporally more extensive observations of covariates Y to improve the spatial and/or temporal continuity of X. ERF accomplishes this scale-aware data fusion, and NS facilitates testing the sensitivity of the data fusion results on different OSDs. In this way NS–ERF makes the power of OSSEs accessible to an entirely new range of use cases. Examples include natural climate solutions (Hemes et al., 2021), emission inventory validation (Desjardins et al., 2018), urban air quality (Vaughan et al., 2021), industry leak detection (Kohnert et al., 2017), and multi-species applications (Vaughan et al., 2017).

### 4.3 Remaining challenges and future work

Notwithstanding these key benefits, an NS–ERF study such as presented here adds labor, computing, and hence funding requirements ahead of the actual field measurements. Considering a typical research grant cycle, one would ideally perform the NS–ERF OSSEs prior to submitting a funding proposal or at least request some level of design flexibility. We conducted the present study over the course of approximately 3 months and utilized the labor and computing resources summarized in Table 4. Overall, we spent ∼ 480 h of labor, or 3 person-months, of which the LES and ERF analyses consumed ∼ 40 % and ∼ 60 %, respectively. The main labor drivers were study conceptualization and setup including data acquisition for LES boundary conditions. It is possible to perform these steps well in advance, e.g., to reduce the NS–ERF effort between grant receipt and field measurements, which is also typically a period with high demand for overall coordination. The 50 h spent on ERF interpretation also included active dialog and iteration with the flight crew, resulting in balanced resources for airborne operation.

Table 4 also shows how LES and ERF differed in their computational needs. LES demanded high-performance computing with 230 000 CPU hours and up to 1.8 TB of memory, which we primarily per-

formed on the US National Center for Atmospheric Research Cheyenne supercomputer (https://w3id.org/smetzger/Metzger-et-al_2021_OSSE-NS-ERF/cheyenne, last access: 29 September 2021). Conversely, ERF required a high-throughput computing architecture, for which we primarily used the US National Science Foundation's CyVerse open science workspace (https://w3id.org/smetzger/Metzger-et-al_2021_OSSE-NS-ERF/cyverse, last access: 29 September 2021). Overall, the strong data requirements of ERF, including use of high-frequency EC data, currently drive NS–ERF computational needs. Investigations into relaxing ERF data requirements while retaining overall performance are in progress. This could permit generating the necessary virtual observations with NSs that substantially reduce resource demand compared to LES, such as closure modeling and Reynolds-averaged Navier–Stokes (e.g., Meyers and Paw U, 1986; Sogachev et al., 2002, 2011; Santiago et al., 2010; Xu et al., 2014). In turn, such modular adjustments promise NS–ERF with reduced complexity and broad accessibility beyond the need for supercomputing, or application to use cases that require consideration of more extensive space, time, and disciplinary domains. Conversely, when designing a natural climate solution (or other) project, NS–ERF could be applied at that project scale, e.g. much smaller or less complex than CHEESEHEAD19, thus reducing overall computational expense. A separate consideration for increasing efficiency could be to further extend the application of value engineering principles, such as an analysis of incremental benefits tapering off with increasing numbers of candidate OSDs. Furthermore, a unified graphical user interface could aid accessibility and usability to better support investigators from diverse backgrounds.

In Sect. 4.1 we discussed several sources of uncertainty that emanated from the specific numerical analyses chosen to optimize the CHEESEHEAD19 flight strategy. In addition, sources of uncertainty that surround the NS–ERF concept as a whole also warrant discussion. One of the strengths of OSSEs in general and NS–ERF in particular is to quantify the efficacy of candidate OSDs for cross-disciplinary applications. However, individual disciplines themselves often invoke very specific criteria and assumptions so their contributions to the overall project are valid (Sect. 1). Also, determining the OSD trade-offs for meeting these discipline-specific requirements could complement NS–ERF with [CE5] an end-to-end science traceability assessment. One direction of future work could use the CHEESEHEAD19 field measurements to derive and evaluate such an end-to-end assessment in general and the presented OSD results in particular.

Furthermore, Earth system observations are highly variable in their space–time extent and resolution (Fig. 1). However, data overlays such as done in ERF require a "least common denominator" space–time resolution among all considered observations. Attaining this least common denominator while retaining quasi-continuous data coverage remains an observational challenge, even for WMO essential climate variables such as land surface temperature. Toolkits that leverage multi-sensor data fusion to provide the necessary resolution and coverage to support plot- to landscape-scale research are only recently emerging (Wu et al., 2013; Pincebourde and Salle, 2020; Desai et al., 2021).

Earth system observations are also variable in their space–time representations. These include gridded remote sensing observations in Eulerian coordinates and EC heat flux observations in Lagrangian coordinates (Metzger, 2018). Data overlays among these observations leverage source area models, which connect the two representations in space and time (Leclerc and Foken, 2014). However, e.g., Bertoldi et al. (2013) and Xu et al. (2020) point out a possible dependency of source area attribution performance on the thermodynamic properties of the quantity observed in Lagrangian coordinates. Robust data overlays across coordinate representations might thus depend on separate source area considerations for neutral density vs. self-buoyant quantities.

## 5   Conclusions

Surface–atmosphere synthesis is traditionally in the vanguard of interdisciplinary research, with efforts ranging from empirical studies over theoretical generalizations to NSs. More recently, data-intensive information discovery promises to further expand our insight into momentum, energy, water, and trace gas cycling. However, "data deluge" rather than the next interdisciplinary breakthrough can result from poor information overlap among ground, airborne, and satellite observations, as well as numerical models. Information gain hinges on our ability to reliably merge information among these perspectives, for which the pre-field stage provides a unique opportunity to optimize the study design accordingly.

We harnessed this opportunity by catalyzing recent advances in conducting virtual experiments within high-resolution NSs ("data from knowledge") and physics-guided data science ("knowledge from data") to create the NS–ERF approach. Traditional data capture focuses on intra-disciplinary best practices, and only in the aftermath does the cross-discipline explanatory power become apparent. In contrast, NS–ERF explores tolerances ("value engineering") in a numerical framework ahead of the actual field deployments, which offers a wide margin for improving cross-discipline post-field synthesis. We used the case study of optimizing the CHEESEHEAD19 OSD as a maiden application for NS–ERF, which maximized the information overlap across micrometeorological and mesometeorological space scales and timescales. To date, these scales have predominantly been dealt with in a discontinuous fashion, which we overcame by combining cross-platform flux tower and flux aircraft responses in a single ERF for the first time. This demonstrated that a carefully designed flight strategy has the potential to double the CHEESEHEAD19 information gain across two

specific design hypotheses and to improve flight operation and crew safety by reducing the number of flight sequences from an originally envisioned 48 to a parsimonious number of 6.

NS–ERF thus makes the benefits of OSSEs accessible for maximizing the information gained from cross-disciplinary field measurements that previously had to rely on experience and expert knowledge alone. This property transcends academic field measurements such as presented here and can inform natural climate solutions, emission inventory validation, urban air quality, industry leak detection, and multi-species applications, among other use cases.

*Code availability.* Software used in this study is available per CHEESEHEAD code policy (https://w3id.org/smetzger/ Metzger-et-al_2021_OSSE-NS-ERF/code-policy, last access: 29 September 2021) and is either already available from or being developed into public code repositories. TS8

*Data availability.* All data used in this study are available per CHEESEHEAD data policy (https://w3id.org/smetzger/ Metzger-et-al_2021_OSSE-NS-ERF/data-policy, last access: 29 September 2021) from the FAIR-compliant CyVerse Data Commons (https://w3id.org/smetzger/Metzger-et-al_2021_ OSSE-NS-ERF/data-commons, Metzger et al., 2021). The top-level document "readme_Metzger-et-al_2021_OSSE-NS-ERF.md" provides specific dataset locations for individual processing steps.

*Author contributions.* SM was responsible for conception and conceptualization. SM, DD, SP, MS, DMP, KX, and ARD developed the methodology. SM, DD, SP, and MS developed software. SM, DD, SP, and MS conducted validation. SM, DD, SP, MS, and KX performed the formal analysis. SM, DD, SP, MS, and ARD conducted the investigation. SM, DD, SP, MS, and ARD acquired resources. SM, DD, and ARD were responsible for data curation. SM, DD, SP, MS, BJB, MM, and LW prepared the original draft. SM, DD, SP, MS, BJB, CF, MM, DMP, LW, and ARD reviewed and edited the manuscript drafts. SM, DD, SP, and BJB created visualization. SM and ARD provided supervision. SM, CF, and ARD were responsible for project administration. SM and ARD acquired funding.

*Competing interests.* The authors declare that they have no conflict of interest.

*Acknowledgements.* This material is based in part upon work supported by the National Science Foundation through the CHEESE-HEAD19 project (grant no. AGS-1822420) and the NEON Program (grant no. DBI-0752017). The National Ecological Observatory Network is a program sponsored by the National Science Foundation and operated under cooperative agreement by Battelle. We thank University of Wyoming King Air pilot Edward Sigel and flight crew for their flight strategy input and James Mineau for creating the flight track maps. Luise Wanner's and Matthias Mauder's contributions were supported by the Deutsche Forschungsgemeinschaft (DFG; grant no. 406980118).

*Financial support.* This research has been supported by the National Science Foundation (grant nos. AGS-1822420 and DBI-0752017) and the Deutsche Forschungsgemeinschaft (grant no. 406980118).

*Review statement.* This paper was edited by Glenn Wolfe and reviewed by two anonymous referees.

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

## Remarks from the language copy-editor

## Remarks from the typesetter