# Peer review of "Novel approach to observing system simulation experiments improves information gain of surface-atmosphere field measurements"

_Atmospheric Measurement Techniques, 2021_

## Author Comment (AC1)

**Author reply to Anonymous Referee #1 comments on the manuscript**

**AMT-2021-86**

**"Observing system simulation experiments double scientific return of surface-atmosphere synthesis"**

by S. Metzger et al.

We thank Anonymous Referee #1 for the valuable feedback, which will substantially improve the manuscript. Please find below the referee comments recited in *blue italics font*, followed by our point-by-point replies in black upright font.

General comments: This is a well-written and interesting analysis of two hours of LES data used to determine optimal aircraft observing strategies for measuring surface latent and sensible heat fluxes when used in combination with a network of surface measurements. One concern is that the recommendations in terms of aircraft track angles and density of tracks is likely to be a function of atmospheric conditions, and the 2 hours analyzed only provide one very limited snapshot of the range of possible conditions. For example, the longitudinal wind streaks present in the LES simulation are known to be a function of wind speed and stability, and in a deep convectively driven boundary layer will not be present. Second, the authors make no mention of clouds, especially boundary layer clouds, which are frequently present in the upper-midwest in summer. Because the manuscript makes no mention of clouds at all, I assume that they did not occur in the simulation. Boundary clouds will certainly change the entire turbulent structure of the ABL. Thus, the results that the authors present are for a very limited range of meteorological conditions. Given the extensive computational resources needed just for this single 2h analysis, it probably was impossible to include these other situations. However, at a minimum, it is essential that the authors point out this limitation to their analysis at the beginning of the manuscript. As currently written, I felt I was misled through most of the manuscript into believing that a more comprehensive analysis of meteorological environments were being evaluated, only to find out later that this was not the case.

**Author reply:**

 Many thanks for sharing these concerns! Indeed we initially aimed to simulate multiple days, as well as morning and afternoon periods. However, as we were racing against the clock to get the actual experiment running, it proved challenging to correctly emulate all candidate observing system designs in all LES runs. As we reached a critical cutoff point we needed to focus on transitioning to the ERF analyses with a single LES run in order to provide numerical results in time for supporting the CHEESEHEAD19 experiment design. With more lead time and computational resources it is possible to realize more LES runs.

- In front of that background, we chose to focus on an LES run that is both typical for the region/season and one that likely generates significant heterogeneity (without the added expense of dealing with clouds). Further considering that for safety reasons the real-world flights were to take place on mostly cloud-free days, we believe the selected case provides a useful case study for how much airborne flight leg choices influence our ability to address CHEESEHEAD19 science objectives.
- While studies with multiple LES runs are available in the literature (e.g., Sühring et al., 2018), it does not appear untypical for case studies to encompass a single LES run, either (e.g., Xu et al., 2020).

Proposed changes to the manuscript:

- Clearer separation throughout the manuscript between (i) the novel, general Numerical Simulation Environmental Response Function (NS-ERF) approach to OSSEs and (ii) the specific CHEESEHEAD 19 airborne case study in the form of LES-ERF.
- Mention in the Abstract, Introduction, Materials and methods, and Discussion that the CHEESHEAD19 case study is limited to two hours of LES data with a single meteorological setting and omission of clouds. Include the specific limitations outlined by the reviewer in the Discussion section.

**Line 99. Please describe what a dispersive flux is.**

Author reply: Dispersive fluxes refer to the scalar transported by standing eddies or spatially organized time-invariant convection cells (e.g., Margairaz et al., 2020; Raupach and Shaw, 1982).

Proposed changes to the manuscript: Add above description of dispersive fluxes.

**Fig. 3c. What do the white areas in the figure denote?**

Author reply: Sect. 2.5 Environmental Response Functions explains: "We also limited the ERF projection to interpolate but not extrapolate outputs, i.e. to only populate grid cells with driver combinations in the range of the virtual measurements. By doing so, the resulting data coverage lets us directly estimate how well we sampled the domain for upscaling."

Proposed changes to the manuscript: Add corresponding explanation to Fig. 3c caption.

**Line 224. What is meant by "super-sampled"?**

Author reply: We were trying to colloquially describe "observing system designs".

Proposed changes to the manuscript: Contextualize "super-sampled" with "13 different observing system designs".

*Line 225. How were the candidate OSDs determined? How was the number of such OSDs to be used determined?*

Author reply: The subsequent Sect. 2.3 Design hypotheses and candidate observing system designs details the rationale for creating the OSDs.

Proposed changes to the manuscript: Insert cross-reference to Sect. 2.3.

Line 231. What are the conditions that prevent ERF from providing a result at a given location?

Author reply: Sect. 2.5 Environmental Response Functions explains: "We also limited the ERF projection to interpolate but not extrapolate outputs, i.e. to only populate grid cells with driver combinations in the range of the virtual measurements. By doing so, the resulting data coverage lets us directly estimate how well we sampled the domain for upscaling."

Proposed changes to the manuscript: Add explanation and cross-reference "...the percentage of grid cells across the study domain that ERF was able to reconstruct within the range of the virtual driver measurements (Sect. 2.5)".

*Line 234. How are the area fluxes determined when there are missing cell data in the ERF domain? Is the ERF spatial average just the average of those cells that have data?*

Author reply: That is correct.

Proposed changes to the manuscript: add clarification "...*horizontal average over all reconstructed grid cells*...".

Line 236. Is the single score an average for all meteorological conditions? For example, I would suspect that the optimal flight tracks might be very different for days with boundary layer cumulus versus clear sky, or early morning/late afternoon transition times compared to midday. (OK, later I see that only one 2 hour mid-day period has been analyzed. It would be helpful to the readers if this was mentioned earlier in the analysis, even in the abstract).

Author reply: Agreed, please see our responses to above general referee comments.

Proposed changes to the manuscript: Please see our proposed changes in response to above general referee comments.

**Line 282. Were there clouds on this simulated day?**

Author reply: No, please see our responses to above general referee comments.

Proposed changes to the manuscript: Please see our proposed changes in response to above general referee comments.

Section 2.5. Lots of complex details of the methodology are given here, but what I do not see discussed in general terms is how one defines the aircraft fluxes (usually an average along one or more flight legs) and then incorporates that data to be able to derive highly resolved spatial maps of the fluxes. A couple of sentences describing the basic principles behind this methodology at the start of the section would be beneficial for readers who are not experts in the technique.

Author reply: Thanks for pointing out the need for providing a brief overview!

Proposed changes to the manuscript: From line 312 add a new overview paragraph with a sequential outline of the methodological steps. Point out principle differences to traditional tower flux (here 1 min vs. traditional 30 min resolution) and airborne flux (here 100 m vs. traditional ~10 km resolution) data processing. Describe how these differences facilitate joining tower and airborne fluxes into space- and time-aligned datasets for machine learning and subsequent flux map projection.

Line 305. A couple of additional sentences here describing the ERF methodology would be useful. If length of the manuscript is a limitation, I would suggest removing some of the philosophical discussion in the introduction in to order to leave some room here.

Author reply: Agreed, please see our response to above referee comments on Sect. 2.5.

Proposed changes to the manuscript: Please see our proposed changes in response to above referee comments on Sect. 2.5.

**Line 351. How was the number of 13 OSDs determined? Was it limited solely by computer resources, or was there any analysis of incremental benefits tapering off with increasing numbers of OSDs?**

Author reply: Sect. 2.3 Design hypotheses and candidate observing system designs details the rationale for creating the OSDs. A value engineering analysis to determine an optimal number of OSDs was not performed. Instead, we created a sample of OSDs that symmetrically covers the trade space presented by the two design hypotheses, and was bracketed by time and computational limitations. Within these limitations, we believe the sample of OSDs provides a useful case study for how much airborne flight leg choices influence our ability to address CHEESEHEAD19 science objectives. We endorse the suggestion of a value-engineering preceding the creation of OSDs, and with more lead time and computational resources it is also possible to realize additional OSDs.

Proposed changes to the manuscript: Add the suggestion of a value-engineering analysis preceding the creation of OSDs to Sect. 4.3 Remaining challenges and future work.

Line 370. The noontime value of 400m for the ABL depth seems surprisingly low. In retrospect, is the day that was simulated here representative of the boundary layers actually observed during the field program? And how was this particular day selected for the analysis? (OK, I

see on line 615 that this is due to an error made in the initialization of LES humidity profile. This error should be mentioned briefly here on line 370, so that readers immediately understand the reason for the low ABL height, instead of wasting time wondering about it while working through the rest of the manuscript).

Author reply: Agreed, thanks for the pointer!

Proposed changes to the manuscript: Add cross-reference to Sect. 4.1 where the issue is discussed in detail.

**Line 434. I am surprised that the range of LST in figure 10a is only 0.1K! Is this range meant to reflect the actual range of LSTs over the 10x10km domain?**

Author reply: Sect. 2.5 Environmental Response Functions explains "*The source area weights provided a linkage* between the sensible and latent heat flux responses in the atmosphere and their spatially resolved drivers at the LES surface (available energy as a proxy for net radiation) and in the first vertical LES layer (land surface temperature and moisture as a proxy for remote-sensing observations)." So while our LST proxy from the first vertical LES layer (10 m resolution) retains the horizontal spatial patterning (25 m resolution), the amplitude of this near-surface air temperature can be expected to be reduced compared to actual surface temperature. Actual surface temperature was not available from the LES run because fluxes were prescribed directly as lower boundary condition in lieu of a land surface model. Furthermore, we constructed ERF response surfaces from source-area-averaged land surface temperature and moisture. This further reduced their amplitude and also explains some of the white gaps in Fig. 3c.

Proposed changes to the manuscript:

Following the suggestion from referee 2, change "land surface temperature (LST)" to "near-surface temperature (NST)", and "land surface moisture (LSM)" to "near-surface moisture (NSM)". Clarify figure caption: "*Example ERF response surfaces.* (a) Sensible heat flux as a function of source-area-averaged energy input and near-surface temperature (NST from the first vertical LES layer, Sect. 2.5). (b) Latent heat flux as a function of source-area-averaged energy input and near-surface moisture (NSM from the first vertical LES layer, Sect. 2.5). For this visualization, all other drivers are kept at their median value." Add a paragraph to Sect. 4.1 Optimizing the CHEESEHEAD19 observing system design: discuss the amplitude ramifications of using first vertical LES level LST and LSM, as well as source area averaging and potential solutions for future applications.

Line 519. The phrase "doubling the scientific return" seems a little grandiose. A more accurate statement would be something along the lines of doubling the accuracy of the spatial sensible and latent heat flux estimates. The scientific return of the measurements taken during the CHEESEHEAD field campaign will be determined many years down the road when all of the analyses of the data set are completed and the papers published. In addition, in view of the

fact that the analysis covered only one particular meteorological condition, "doubling the scientific return" seems really an exaggeration.

**Author reply:**

- We thank the reviewer for this observation and agree that in particular when taken out of context the term "scientific return" in itself could be interpreted as grandiose, which was not our intent.
- We feel there might be two common root causes for this and a similar concern by referee #2. First, the current manuscript structure lacks separation between (i) the novel, general NS-ERF approach to OSSEs and (ii) its maiden, example application to the CHEESEHEAD19 airborne case study in the form of the specific LES-ERF realization. Second, we introduce NS-ERF-specific terminology in the text (e.g., optimality criterion, hypothesis, objective, observing system design, **scientific return** etc.). However, we do not currently provide a clear conceptual overview of the relationships among these terms, thus creating a lack of context for interpreting the CHEESEHEAD19 case study results.

Proposed changes to the manuscript:

- Replace "scientific return" with "information gain" throughout the manuscript.
- Re-organize manuscript to more clearly distinguish between (i) the novel NS-ERF approach to OSSEs and (ii) its maiden, example application to the CHEESEHEAD19 airborne case study in the specific form of LES-ERF.
- Reflect this shift by rewording the title to "Novel approach to observing system simulation experiments improves information gain of surface-atmosphere field measurements". Specifically, shift title emphasis from case study results (replace "...double scientific return..." with "improves information gain of surface-atmosphere field measurements") to creating an extensible OSSE framework for designing surface-atmosphere observing systems (add "novel approach").
- Reword Sect. 2.2 header to "NS-ERF observing system simulation experiments". Add a flowchart/"visual glossary" to the beginning of the section for inter-relating generalized NS-ERF components/terminology including the aim to maximize information gain. At the end of the section (following "We then used the arithmetic mean and standard deviation to aggregate CR1–CR3 across flight patterns, flight sequences, and ultimately among themselves into a single score (Sect. 3.3)."): define information gain specifically for the CHEESEHEAD19 case study as the arithmetic mean of three optimality criteria that directly correspond to CHEESEHEAD19 science objectives.
- In above Sect. 2.2 and discussion Sect. 4.1 Optimizing the CHEESEHEAD19 observing system design: clearly distinguish the potential to improve the information gain of CHEESEHEAD19 airborne measurements as evidenced by the 2-hour case study from the actual scientific return of the measurements taken during the CHEESEHEAD field campaign.

**Line 678. "und" should be "and".**

Author reply: Thanks for catching this.

Proposed changes to the manuscript: Correct spelling.

**References**

Margairaz, F., Pardyjak, E. R., and Calaf, M.: Surface thermal heterogeneities and the atmospheric boundary layer: The relevance of dispersive fluxes, Boundary Layer Meteorol., 175, 369-395, doi:10.1007/s10546-020-00509-w, 2020.

Raupach, M. R., and Shaw, R. H.: Averaging procedures for flow within vegetation canopies, Boundary Layer Meteorol., 22, 79-90, doi:10.1007/BF00128057, 1982.

Sühring, M., Metzger, S., Xu, K., Durden, D., and Desai, A.: Trade-offs in flux disaggregation: a large-eddy simulation study, Boundary Layer Meteorol., 170, 69-93, doi:10.1007/s10546-018-0387-x, 2018.

Xu, K., Sühring, M., Metzger, S., Durden, D., and Desai, A. R.: Can data mining help eddy covariance see the landscape? A large-eddy simulation study, Boundary Layer Meteorol., Online First, doi:10.1007/s10546-020-00513-0, 2020.

---

## Author Comment (AC2)

**"Observing system simulation experiments double scientific return of surface-atmosphere synthesis"**

**by S. Metzger et al.**

We thank Anonymous Referee #2 for the valuable feedback, which will substantially improve the manuscript. Please find below the referee comments recited in *blue italics font*, followed by our point-by-point replies in black upright font.

*General comments*

*Metzger et al present an article where they describe the importance of using observing system designs (OSD) to maximize scientific insights from surface air exchange measurements. The authors propose that one particular observation system design that is useful for the study of biosphere-atmosphere interactions is the Observation System Simulation Experiment (OSSE), which is obtained through the use of Large Eddy Simulations (LES) in combination with Environmental Response Functions as previously described by Metzger (2013). Thus, the OSSE used in this study is referred as LES_ERF..*

*The authors argue that most researchers in the field set up the instrumentation first to gain "knowledge from data", however, an alternative is to use OSDs such as this OSSE to gain "data from knowledge" and thus optimize the scientific insight gained from the observations. For example, this type of optimization can be used to determine an optimal location for an eddy-covariance sensor both vertically, and horizontally, thus guaranteeing the adequate fetch but also guaranteeing the correct spatial heterogeneity that may account for mesoscale circulations.*

*The goal of the paper is to show that making informed Observing System Designs (OSDs) for surface-atmosphere field measurements can improve the amount of useful information, or as the authors say, "double the scientific return". And that creating OSSE can be one of the best ways to attain an optimal OSD. The novel approach is to provide design information prior to testing OSDs in the field.*

*The authors make use of the CHEESSEHEAD19 dataset, which was originally designed to test the hypothesis that mesoscale features, driven by surface heterogeneity can explain the lack of closure in energy balance. The campaign was also used to evaluate the use of Environmental Response Functions (ERFs) for estimating "fluxes in a box" an approach previously created by the authors. In this paper, the authors use LES_ERF to find the optimal flight strategy to*

*maximize the amount of useful information to evaluate the energy balance closure at the CHESSEHEAD19 site. The OSDs are built around 2 hypotheses, one is that it is critical for airborne EC to measure perpendicular to the prevailing wind, and 2, that it is more informative to fly a finely spaced pattern. 13 different OSD were created: the first based on tower-only data, and the other 12 based on the combination of four track angles (0,45,60,90 °), and 3 flight patterns (Alternating, Outbound, Return). The results show that flight patterns with a track angle of 45-90 double the percent improvement based on three optimality criteria when compared to parallel flight patterns.*

> Author reply: Many thanks for this excellent summary.

*I think the paper addresses relevant scientific questions within the scope of Atmospheric Measurements Techniques, and that it presents novel concepts, ideas, and tools. However, I think there are some issues with the clarity of the methodology and with the application of this technique to other studies.*

> Author reply: Many thanks for sharing these concerns, which we address individually below.

*The methodology can be very confusing. This is what I understood from section 2.2. The methodology consists of first combining information from different sources into ERFs. This would be, for example, combining information from existing towers and aircraft data to create a space-and-time aligned dataset. The second step is to then create multiple OSDs by using LES, and finally benchmark the candidate OSDs by evaluating if the candidate OSD can recreate the environment from the original ERF. One of the 3 ways to evaluate this approach, is the Energy Balance Ratio, defined as the sum of the sensible and latent heat fluxes produced by the ERF, to the sum of latent and sensible heat fluxes produced by the LES. It is my understanding that the numerator in equation 1 does not vary and that the denominator should vary according to different OSDs. But I may be wrong. This section could use some synthesis to make it clearer to the reader. I think there are some conflicts between what is said in the caption and what is said in section 2.2.*

> Author reply:
> - Many thanks for pointing out apparent inconsistencies in Sect. 2.2. We believe that these arise primarily from two ambiguities:
> - First, Figure 3 initially introduces ERF (Figure 3a) and then LES (Figure 3b), rather than following the actual sequence of LES-ERF data processing (first LES, then ERF).
> - Second, we directly prescribed time-dependent, heterogeneous sensible and latent heat flux grids as LES surface forcings. For this we utilized the pre-existing sensible and latent heat flux grids of Metzger et al. (2013). Metzger et al. (2013) coincidentally used ERF and observational data to derive these grids in their independent study, though any other appropriate dataset or methodology could have been used to provide the LES surface forcings. The

Metzger et al. (2013) grids were merely a suitable dataset accessible to us at the time, irrespective of methodology.

Proposed changes to the manuscript:

- Reverse the sequence of Figure 3a and Figure 3b to follow the actual sequence of LES-ERF data processing (first LES, then ERF).
- Remove any notion of ERF when citing the use of the Metzger et al. (2013) sensible and latent heat flux grids, to forego conflation with LES-ERF.
- In Sect. 2 Materials and methods add a paragraph that links the introduction to a "big picture" overview of the subsequent methods subsections. Distinguish between (i) the novel NS-ERF approach to OSSEs and (ii) its maiden, example application to the CHEESEHEAD19 case study in the form of LES-ERF.
- In the beginning of Sect. 2.2, add a flowchart of the generalized NS-ERF approach that acts as a "visual glossary" by inter-relating individual components/terminology (e.g., optimality criterion, hypothesis, objective, observing system design, scientific return etc.). Explain how these generalized components are then specifically realized in the CHEESEHEAD19 case study in the form of LES-ERF.
- Explain that the numerator in Eq. 1 does not vary and that the denominator varies according to the different OSDs.

*After reading the results (L. 440) I now see that there is a baseline OSD for the tower-only dataset and 12 other OSDs for the combined tower-aircraft. The tower-only dataset has a given spatial coverage that can be improved by aircraft sampling, and the goal is to maximize this coverage by deciding on certain flight tracks. By applying different OSDs with different flight angles, it was found that the spatial coverage is maximized in a perpendicular flight track (25% improvement). In table 1, similar analyses are given for energy balance ratio, and spatial patterning, the other two optimality criteria.*

Author reply: Many thanks for this excellent summary! These relationships are also introduced in Sect. 2.3 l. 269ff.: "*Based on this super-sample we evaluated 13 candidate OSDs. Applying LES-ERF to 44 site-hours of data from the virtual EC tower network alone provided the **baseline OSD**. Combining data from the virtual EC tower network with one of the 3 flight sequences × 4 flight patterns = 12 airborne EC combinations provided 12 **alternative OSDs**. Each of the alternative OSDs consisted of 44 site-hours virtual tower EC data and 11 flight tracks × 25 km = 275 km virtual airborne EC data. This configuration allows us to **evaluate the change in the optimality criteria (Sect. 2.2) for each of the 12 joint tower and aircraft alternative OSDs relative to the tower-only baseline OSD**.*".

Proposed changes to the manuscript: Add the referee's "plain language summary" to above paragraph in Sect. 2.3.

*Reading section 2.4, I see that the LES are created with surface fields of H and LE created from the ERF. So, isn't this some circular reasoning? You are creating OSDs from ERFs that are benchmarked against LES created using ERF… Please explain*

Author reply: Please see our responses to above referee comment "*The methodology can be very confusing…*". We believe that the present comment originates from the same two ambiguities detailed in our response above.

Proposed changes to the manuscript: Please see our proposed changes in response to above referee comment "*The methodology can be very confusing…*".

*I think the results are clear and easy to follow but as mentioned previously the methodology can be a little confusing and I think needs more synthesis and needs to be clearer.*

Author reply: Please see our responses to above referee comment "*The methodology can be very confusing…*". We believe that the present comment originates from the same two ambiguities detailed in our response above.

Proposed changes to the manuscript: Please see our proposed changes in response to above referee comment "*The methodology can be very confusing…*".

*It is my understanding that the LES_ERF approach is designed specifically for the combination of airborne EC measurements with tower EC measurements. If this is the case this needs to be stated clearly in the abstract and perhaps even in the title.*

Author reply: Many thanks for sharing this observation! The presented approach to OSSEs is indeed widely applicable beyond combining airborne EC measurements with tower EC measurements. We feel that the impression of limited applicability likely originates from the current manuscript structure lacking separation between (i) the novel, general Numerical Simulation - Environmental Response Function (NS-ERF) approach to OSSEs and (ii) its maiden, example application to the CHEESEHEAD19 airborne case study in the form of the specific LES-ERF realization of NS-ERF. Furthermore, the manuscript does not currently specify which alternative form NS-ERF components could take for natural climate solutions, emission inventory validation, urban air quality, industry leak detection, and multi-species applications.

Proposed changes to the manuscript:

- Re-organize manuscript to more clearly distinguish between (i) the novel NS-ERF approach to OSSEs and (ii) its maiden, example application to the CHEESEHEAD19 airborne case study in the specific form of LES-ERF.
- Reword Sect. 2.2 header to "NS-ERF observing system simulation experiments". Add a flowchart/"visual glossary" to the beginning of the section for inter-relating generalized NS-ERF components/terminology.
- Substantiate extensibility claim by adding concrete examples for case study substitution where use cases are mentioned throughout the manuscript. In the discussion Sect. 4.2, spell out how to extend to any sort of study where

measurements of X across space and time need to be combined to estimate some surface or atmospheric property Y using input covariates Z in ERF, as simulated in NS.

*To me, it is not clear what the "scientific return" means. How can you double the scientific return? Isn't this all subjective? Whatever information you gained by using one flight path instead of the other depends on how you interpret it. Doesn't it? What the results show is that flight patterns with a track angle of 45-90 double the percent improvement based on three optimality criteria, when compared to parallel flight patterns. I'm also not sure how you can "order-of-magnitude improve flight operation and crew safety". The last two statements are part of the main conclusion but the way they are quantified seems subjective.*

Author reply:

- We agree with the referee's sentiment that "*Whatever information you gained by using one flight path instead of the other depends on how you interpret it*".
- OSSEs in general aim to provide an objective interpretation of "information gain" based on quantifying the ability to address the science objectives of a given application. To systematically realize this aim for surface-atmosphere field measurements we created the NS-ERF approach to OSSEs, and provide a case study example (CHEESEHEAD19 airborne design in the form of LES-ERF).
- In Sects. 2.2 and 2.3 we introduce the relevant NS-ERF components and their terminology in the text (e.g., optimality criterion, hypothesis, objective, observing system design, **scientific return** etc.). However, we do not currently provide a clear conceptual overview of the relationships among these components, thus creating a lack of context for objectively interpreting the CHEESEHEAD19 case study results.
- Sect. 4.1 specifies: "*On the other hand, the parsimonious number of only 6 flight sequences and an even smaller number of 3 sets of waypoints simplify flight planning and navigation. In combination with entirely avoiding the town and airfield of Park Falls this promotes flight crew safety by an order of magnitude compared to the originally envisioned 48 flight sequences. Specifically, it frees up the flight crew from arduous navigation details, thus reducing fatigue, increasing awareness during the 100 m low-level flight maneuvers, and ultimately reducing the margin for human error.*"

Proposed changes to the manuscript:

- Replace "scientific return" with "information gain" throughout the manuscript.
- Reword the title to "Novel approach to observing system simulation experiments improves information gain of surface-atmosphere field measurements".
- Reword Sect. 2.2 header to "NS-ERF observing system simulation experiments". Add a flowchart/"visual glossary" to the beginning of Sect. 2.2 for inter-relating NS-ERF components/terminology including the aim to

maximize information gain. At the end of the section (following "*We then used the arithmetic mean and standard deviation to aggregate CR1–CR3 across flight patterns, flight sequences, and ultimately among themselves into a single score (Sect. 3.3).*"): define information gain specifically for the CHEESEHEAD19 case study as the arithmetic mean of three optimality criteria that directly correspond to CHEESEHEAD19 science objectives.

- In Sect. 5 Conclusion, replace *"order-of-magnitude improve flight operation and crew safety"* with "improve flight operation and crew safety by reducing the originally envisioned 48 flight sequences to a parsimonious number of only 6 flight sequences".

*Lastly, a question of applicability. How many other researchers are in the capacity to apply airborne EC measurements with such large-scale deployment of towers and the capacity to run computationally expensive LES at this scale? The authors present a good analysis of other large-scale field campaigns in section 4.2 but still, I'm not sure about the use of this approach to support the last conclusion of the abstract that "the approach lends itself to optimize observing system designs also for natural climate solutions, emission inventory validation, urban air quality, industry leak detection, and multi-species applications" What would be the cost-benefit analysis of implementing a large-scale field campaign like this for every natural climate solution project?*

Author reply:

- Many thanks for raising this point. The presented NS-ERF approach to OSSEs is applicable to field measurements in general, and large-scale deployments or even an airborne component are by no means a requirement. For additional detail please see our responses to the above referee comment "*It is my understanding that the LES_ERF approach is designed specifically for the combination of airborne EC measurements with tower EC measurements…*"
- So when designing a natural climate solution (or other) project, NS-ERF could be applied at that project scale, e.g. much smaller compared to CHEESEHEAD19. Furthermore, an alternate NS-ERF configuration could be considered such as replacing LES (from LES-ERF) with RANS (to RANS-ERF) as discussed in Sect. 4.3. Such modular adjustments and scalability reduce computational expense and makes the general NS-ERF approach to OSSEs accessible to a wide range of applications.
- The list of large-scale field campaigns in the manuscript are meant to provide context for the CHEESEHEAD19 airborne case study.

Proposed changes to the manuscript:

- Please see our proposed changes in response to above referee comment "*It is my understanding that the LES_ERF approach is designed specifically for the combination of airborne EC measurements with tower EC measurements…*"
- Specify that the NS-ERF approach to OSSEs is applicable to field measurements in general, and large-scale deployments or even an airborne component are by no means a requirement.

- Expand concrete examples for case study substitution with scalability considerations incl. project size, NS-ERF configuration options, etc.

*Specific comments*

*31 This approach doubled the ability to explore energy balance closure? This is a very subjective statement. How do you double the ability to explore?*

Author reply: Thanks for this pointer. Please see our responses to above referee comment "*To me, it is not clear what the "scientific return" means…*". We believe that the present comment originates from the same ambiguity detailed in our response above.

Proposed changes to the manuscript: Please see our proposed changes in response to above referee comment "*To me, it is not clear what the "scientific return" means…*". Furthermore, change "*…doubled CHEESEHEAD19's ability…*" to "doubled CHEESEHEAD19's potential" throughout the manuscript.

*371. Please specify the subsection in Section 4 where we can learn why the PBL was so low*

Author reply: Thanks for the pointer!

Proposed changes to the manuscript: Add cross-reference to Sect. 4.1 where the issue is discussed in detail.

*409 Should this be "near-surface moisture"? "Land surface moisture" sounds like soil moisture and is redundant.*

Author reply: Thanks for this pointer.

Proposed changes to the manuscript: Change "land surface moisture (LSM)" to "near-surface moisture (NSM)" and "land surface temperature (LST)" to "near-surface temperature (NST)".

*L.465 Indicate in the table that these are percent values*

Author reply: Many thanks for your suggestion.

Proposed changes to the manuscript: Update Table 1 – Table 3 accordingly.

*I find that in the introduction and methods there are multiple statements that are hard to follow, such as:*

*192 "Here, we propose the extensible LES-ERF approach that explicitly simulates the joint scientific return in response to different candidate OSDs for addressing user-defined design hypotheses"*

*I think part of the problem is the use of the term "scientific return", which should be reevaluated.*

Author reply: Thanks for this pointer. Please see our responses to above referee comment "*To me, it is not clear what the "scientific return" means…*". We believe that the present comment originates from the same ambiguity detailed in our response above.

Proposed changes to the manuscript: Replace "scientific return" with "information gain" throughout the manuscript. For additional detail please also see our proposed changes in response to above referee comment "*To me, it is not clear what the "scientific return" means…*".

*I also think that an effort to synthesize the introduction and the methods would help the readability of the paper.*

Author reply: Agreed, many thanks for this observation!

Proposed changes to the manuscript: In Sect. 2 Materials and methods add a paragraph that links the introduction to a "big picture" overview of the subsequent methods subsections. Distinguish between (i) the novel NS-ERF approach to OSSEs and (ii) its maiden, example application to the CHEESEHEAD19 case study in the form of LES-ERF.

*L.286 What is the LES time step?*

Author reply: The LES time step was set at 0.4 s for the analysis period. Mentioned in line 303.

*Technical comments*

*Should the verb be "creating" rather than "observing". Aren't you creating these OSSE?*

Author reply: We were not able to locate this comment in the manuscript.

**References**

Metzger, S., Xu, K., Desai, A. R., Taylor, J. R., Kljun, N., Schneider, D., Kampe, T., and Fox, A.: Spatio-temporal rectification of tower-based eddy-covariance flux measurements for consistently informing process-based models, 46th AGU annual Fall Meeting, San Francisco, U.S.A., 9 - 13 December, 2013.

---

## Author Response (AR1)

**"Observing system simulation experiments double scientific return of surface-atmosphere synthesis"**

**by S. Metzger et al.**

We thank Anonymous Referee #1 for the valuable feedback, which will substantially improve the manuscript. Please find below the referee comments recited in *blue italics font*, followed by our point-by-point replies in black upright font.

*General comments: This is a well-written and interesting analysis of two hours of LES data used to determine optimal aircraft observing strategies for measuring surface latent and sensible heat fluxes when used in combination with a network of surface measurements. One concern is that the recommendations in terms of aircraft track angles and density of tracks is likely to be a function of atmospheric conditions, and the 2 hours analyzed only provide one very limited snapshot of the range of possible conditions. For example, the longitudinal wind streaks present in the LES simulation are known to be a function of wind speed and stability, and in a deep convectively driven boundary layer will not be present. Second, the authors make no mention of clouds, especially boundary layer clouds, which are frequently present in the upper-midwest in summer. Because the manuscript makes no mention of clouds at all, I assume that they did not occur in the simulation. Boundary clouds will certainly change the entire turbulent structure of the ABL. Thus, the results that the authors present are for a very limited range of meteorological conditions. Given the extensive computational resources needed just for this single 2h analysis, it probably was impossible to include these other situations. However, at a minimum, it is essential that the authors point out this limitation to their analysis at the beginning of the manuscript. As currently written, I felt I was misled through most of the manuscript into believing that a more comprehensive analysis of meteorological environments were being evaluated, only to find out later that this was not the case.*

Author reply:

- Many thanks for sharing these concerns! Indeed we initially aimed to simulate multiple days, as well as morning and afternoon periods. However, as we were racing against the clock to get the actual experiment running, it proved challenging to correctly emulate all candidate observing system designs in all LES runs. As we reached a critical cutoff point we needed to focus on transitioning to the ERF analyses with a single LES run in order to provide numerical results in time for supporting the CHEESEHEAD19 experiment

design. With more lead time and computational resources it is possible to realize more LES runs.

- In front of that background, we chose to focus on an LES run that is both typical for the region/season and one that likely generates significant heterogeneity (without the added expense of dealing with clouds). Further considering that for safety reasons the real-world flights were to take place on mostly cloud-free days, we believe the selected case provides a useful case study for how much airborne flight leg choices influence our ability to address CHEESEHEAD19 science objectives.
- While studies with multiple LES runs are available in the literature (e.g., Sühring et al., 2018), it does not appear untypical for case studies to encompass a single LES run, either (e.g., Xu et al., 2020).

Changes to the manuscript:

- Clearer separation throughout the manuscript between (i) the novel, general Numerical Simulation - Environmental Response Function (NS-ERF) approach to OSSEs and (ii) the specific CHEESEHEAD 19 airborne case study.
- Mentioned in the Abstract, Introduction, Materials and methods, and Discussion that the CHEESEHEAD19 case study is limited to two hours of LES data with a single meteorological setting and omission of clouds. Included the specific limitations outlined by the reviewer in the Discussion section.

*Line 99. Please describe what a dispersive flux is.*

Author reply: Dispersive fluxes refer to the scalar transported by standing eddies or spatially organized time-invariant convection cells (e.g., Margairaz et al., 2020; Raupach and Shaw, 1982).

Changes to the manuscript: Added above description of dispersive fluxes.

*Fig. 3c. What do the white areas in the figure denote?*

Author reply: Sect. 2.5 Environmental Response Functions explains: "*We also limited the ERF projection to interpolate but not extrapolate outputs, i.e. to only populate grid cells with driver combinations in the range of the virtual measurements. By doing so, the resulting data coverage lets us directly estimate how well we sampled the domain for upscaling.*"

Changes to the manuscript: Added corresponding explanation to Fig. 3c caption.

*Line 224. What is meant by "super-sampled"?*

Author reply: We were trying to colloquially describe "observing system designs".

Changes to the manuscript: Contextualized "super-sampled" with "13 different observing system designs".

*Line 225. How were the candidate OSDs determined? How was the number of such OSDs to be used determined?*

Author reply: The subsequent Sect. 2.3 Design hypotheses and candidate observing system designs details the rationale for creating the OSDs.

Changes to the manuscript: Inserted cross-reference to Sect. 2.3.

*Line 231. What are the conditions that prevent ERF from providing a result at a given location?*

Author reply: Sect. 2.5 Environmental Response Functions explains: "*We also limited the ERF projection to interpolate but not extrapolate outputs, i.e. to only populate grid cells with driver combinations in the range of the virtual measurements. By doing so, the resulting data coverage lets us directly estimate how well we sampled the domain for upscaling.*"

Changes to the manuscript: Added explanation and cross-reference "*...the percentage of grid cells across the study domain that ERF was able to reconstruct within the range of the virtual driver measurements (Sect. 2.5)*".

*Line 234. How are the area fluxes determined when there are missing cell data in the ERF domain? Is the ERF spatial average just the average of those cells that have data?*

Author reply: That is correct.

Changes to the manuscript: added clarification "*...horizontal average over all reconstructed grid cells...*".

*Line 236. Is the single score an average for all meteorological conditions? For example, I would suspect that the optimal flight tracks might be very different for days with boundary layer cumulus versus clear sky, or early morning/late afternoon transition times compared to mid-day. (OK, later I see that only one 2 hour mid-day period has been analyzed. It would be helpful to the readers if this was mentioned earlier in the analysis, even in the abstract).*

Author reply: Agreed, please see our responses to above general referee comments.

Changes to the manuscript: Please see our changes in response to above general referee comments.

*Line 282. Were there clouds on this simulated day?*

Author reply: No, please see our responses to above general referee comments.

Changes to the manuscript: Please see our changes in response to above general referee comments.

*Section 2.5. Lots of complex details of the methodology are given here, but what I do not see discussed in general terms is how one defines the aircraft fluxes (usually an average along one or more flight legs) and then incorporates that data to be able to derive highly resolved spatial maps of the fluxes. A couple of sentences describing the basic principles behind this methodology at the start of the section would be beneficial for readers who are not experts in the technique.*

Author reply: Thanks for pointing out the need for providing a brief overview!

Changes to the manuscript: From line 312 added several sentences with a sequential outline of the methodological steps. Pointed out principle differences to traditional tower flux (here 1 min vs. traditional 30 min resolution) and airborne flux (here 100 m vs. traditional ~10 km resolution) data processing. Described how these differences facilitate joining tower and airborne fluxes into space- and time-aligned datasets for machine learning and subsequent flux map projection.

*Line 305. A couple of additional sentences here describing the ERF methodology would be useful. If length of the manuscript is a limitation, I would suggest removing some of the philosophical discussion in the introduction in to order to leave some room here.*

Author reply: Agreed, please see our response to above referee comments on Sect. 2.5.

Changes to the manuscript: Please see our changes in response to above referee comments on Sect. 2.5.

*Line 351. How was the number of 13 OSDs determined? Was it limited solely by computer resources, or was there any analysis of incremental benefits tapering off with increasing numbers of OSDs?*

Author reply: Sect. 2.3 Design hypotheses and candidate observing system designs details the rationale for creating the OSDs. A value engineering analysis to determine an optimal number of OSDs was not performed. Instead, we created a sample of OSDs that symmetrically covers the trade space presented by the two design hypotheses, and was bracketed by time and computational limitations. Within these limitations, we believe the sample of OSDs provides a useful case study for how much airborne flight leg choices influence our ability to address CHEESEHEAD19 science objectives. We endorse the suggestion of a value-engineering preceding the creation of OSDs, and with more lead time and computational resources it is also possible to realize additional OSDs.

Changes to the manuscript: Added the suggestion of a value-engineering analysis preceding the creation of OSDs to Sect. 4.3 Remaining challenges and future work.

*Line 370. The noontime value of 400m for the ABL depth seems surprisingly low. In retrospect, is the day that was simulated here representative of the boundary layers actually observed during the field program? And how was this particular day selected for the analysis? (OK, I see on line 615 that this is due to an error made in the initialization of LES humidity profile.*

*This error should be mentioned briefly here on line 370, so that readers immediately understand the reason for the low ABL height, instead of wasting time wondering about it while working through the rest of the manuscript).*

Author reply: Agreed, thanks for the pointer!

Changes to the manuscript: Added cross-reference to Sect. 4.1 where the issue is discussed in detail.

*Line 434. I am surprised that the range of LST in figure 10a is only 0.1K! Is this range meant to reflect the actual range of LSTs over the 10x10km domain?*

Author reply: Sect. 2.5 Environmental Response Functions explains "***The source area weights provided a linkage** between the sensible and latent heat flux responses in the atmosphere and their spatially resolved drivers at the LES surface (available energy as a proxy for net radiation) and in the **first vertical LES layer (land surface temperature** and moisture as a **proxy for remote-sensing observations**).*" So while our LST proxy from the first vertical LES layer (10 m resolution) retains the horizontal spatial patterning (25 m resolution), the amplitude of this near-surface air temperature can be expected to be reduced compared to actual surface temperature. Actual surface temperature was not available from the LES run because fluxes were prescribed directly as lower boundary condition in lieu of a land surface model. Furthermore, we constructed ERF response surfaces from source-area-averaged land surface temperature and moisture. This further reduced their amplitude and also explains some of the white gaps in Fig. 3c.

Changes to the manuscript:

Following the suggestion from referee 2, changed "land surface temperature (LST)" to "near-surface temperature (NST)", and "land surface moisture (LSM)" to "near-surface moisture (NSM)". Clarified figure caption: "*Example ERF response surfaces. (a) Sensible heat flux as a function of **source-area-averaged** energy input and near-surface temperature (NST **from the first vertical LES layer, Sect. 2.5**). (b) Latent heat flux as a function of **source-area-averaged** energy input and near-surface moisture (NSM **from the first vertical LES layer, Sect. 2.5**). For this visualization, all other drivers are kept at their median value.*" Added two sentence to Sect. 2.5 Environmental Response Functions: "*While these near-surface temperature and moisture retain much of the horizontal spatial patterning, their amplitudes are reduced compared to actual surface temperature and moisture. This is further confounded by the source-area-averaging applied here, and the combined effects on amplitude are evident e.g. in Fig. 10.*"

*Line 519. The phrase "doubling the scientific return" seems a little grandiose. A more accurate statement would be something along the lines of doubling the accuracy of the spatial sensible and latent heat flux estimates. The scientific return of the measurements taken during the CHEESEHEAD field campaign will be determined many years down the road when all of the analyses of the data set are completed and the papers published. In addition, in view of the*

*fact that the analysis covered only one particular meteorological condition, "doubling the scientific return" seems really an exaggeration.*

Author reply:

- We thank the reviewer for this observation and agree that – in particular when taken out of context – the term "scientific return" in itself could be interpreted as grandiose, which was not our intent.
- We feel there might be two common root causes for this and a similar concern by referee #2. First, the current manuscript structure lacks separation between (i) the novel, general NS-ERF approach to OSSEs and (ii) its maiden, example application to the CHEESEHEAD19 airborne case study. Second, we introduce NS-ERF-specific terminology in the text (e.g., optimality criterion, hypothesis, objective, observing system design, **scientific return** etc.). However, we do not currently provide a clear conceptual overview of the relationships among these terms, thus creating a lack of context for interpreting the CHEESEHEAD19 case study results.

Changes to the manuscript:

- Replaced "scientific return" with "information gain" throughout the manuscript.
- Re-organize manuscript to more clearly distinguish between (i) the novel NS-ERF approach to OSSEs and (ii) its maiden, example application to the CHEESEHEAD19 airborne case study.
- Reflected this shift by rewording the title to "Novel approach to observing system simulation experiments improves information gain of surface-atmosphere field measurements". Specifically, shifted title emphasis from case study results (replace "*...double scientific return...*" with "improves information gain of surface-atmosphere field measurements") to creating an extensible OSSE framework for designing surface-atmosphere observing systems (add "novel approach").
- Reworded Sect. 2.2 header to "NS-ERF observing system simulation experiments". Added a flowchart/"visual glossary" to the beginning of the section for inter-relating generalized NS-ERF components/terminology including the aim to maximize information gain. At the end of the section (following "*We then used the arithmetic mean and standard deviation to aggregate CR1–CR3 across flight patterns, flight sequences, and ultimately among themselves into a single score (Sect. 3.3).*"): defined information gain specifically for the CHEESEHEAD19 case study as the arithmetic mean of three optimality criteria that directly correspond to CHEESEHEAD19 science objectives.
- In above Sect. 2.2 and discussion Sect. 4.1 Optimizing the CHEESEHEAD19 observing system design: added distinction between the potential to improve the information gain of CHEESEHEAD19 airborne measurements as evidenced by the 2-hour case study and the actual scientific return of the measurements taken during the CHEESEHEAD field campaign.

*Line 678. "und" should be "and".*

Author reply: Thanks for catching this.

Changes to the manuscript: Corrected spelling.

**"Observing system simulation experiments double scientific return of surface-atmosphere synthesis"**

**by S. Metzger et al.**

We thank Anonymous Referee #2 for the valuable feedback, which will substantially improve the manuscript. Please find below the referee comments recited in *blue italics font*, followed by our point-by-point replies in black upright font.

*General comments*

*Metzger et al present an article where they describe the importance of using observing system designs (OSD) to maximize scientific insights from surface air exchange measurements. The authors propose that one particular observation system design that is useful for the study of biosphere-atmosphere interactions is the Observation System Simulation Experiment (OSSE), which is obtained through the use of Large Eddy Simulations (LES) in combination with Environmental Response Functions as previously described by Metzger (2013). Thus, the OSSE used in this study is referred as LES_ERF..*

*The authors argue that most researchers in the field set up the instrumentation first to gain "knowledge from data", however, an alternative is to use OSDs such as this OSSE to gain "data from knowledge" and thus optimize the scientific insight gained from the observations. For example, this type of optimization can be used to determine an optimal location for an eddy-covariance sensor both vertically, and horizontally, thus guaranteeing the adequate fetch but also guaranteeing the correct spatial heterogeneity that may account for mesoscale circulations.*

*The goal of the paper is to show that making informed Observing System Designs (OSDs) for surface-atmosphere field measurements can improve the amount of useful information, or as the authors say, "double the scientific return". And that creating OSSE can be one of the best ways to attain an optimal OSD. The novel approach is to provide design information prior to testing OSDs in the field.*

*The authors make use of the CHEESSEHEAD19 dataset, which was originally designed to test the hypothesis that mesoscale features, driven by surface heterogeneity can explain the lack of closure in energy balance. The campaign was also used to evaluate the use of Environmental Response Functions (ERFs) for estimating "fluxes in a box" an approach previously created by the authors. In this paper, the authors use LES_ERF to find the optimal flight strategy to maximize the amount of useful information to evaluate the energy balance closure at the*

*CHESSEHEAD19 site. The OSDs are built around 2 hypotheses, one is that it is critical for airborne EC to measure perpendicular to the prevailing wind, and 2, that it is more informative to fly a finely spaced pattern. 13 different OSD were created: the first based on tower-only data, and the other 12 based on the combination of four track angles (0,45,60,90 °), and 3 flight patterns (Alternating, Outbound, Return). The results show that flight patterns with a track angle of 45-90 double the percent improvement based on three optimality criteria when compared to parallel flight patterns.*

> Author reply: Many thanks for this excellent summary.

*I think the paper addresses relevant scientific questions within the scope of Atmospheric Measurements Techniques, and that it presents novel concepts, ideas, and tools. However, I think there are some issues with the clarity of the methodology and with the application of this technique to other studies.*

> Author reply: Many thanks for sharing these concerns, which we address individually below.

*The methodology can be very confusing. This is what I understood from section 2.2. The methodology consists of first combining information from different sources into ERFs. This would be, for example, combining information from existing towers and aircraft data to create a space-and-time aligned dataset. The second step is to then create multiple OSDs by using LES, and finally benchmark the candidate OSDs by evaluating if the candidate OSD can recreate the environment from the original ERF. One of the 3 ways to evaluate this approach, is the Energy Balance Ratio, defined as the sum of the sensible and latent heat fluxes produced by the ERF, to the sum of latent and sensible heat fluxes produced by the LES. It is my understanding that the numerator in equation 1 does not vary and that the denominator should vary according to different OSDs. But I may be wrong. This section could use some synthesis to make it clearer to the reader. I think there are some conflicts between what is said in the caption and what is said in section 2.2.*

> Author reply:
> - Many thanks for pointing out apparent inconsistencies in Sect. 2.2. We believe that these arise primarily from two ambiguities:
> - First, Figure 3 initially introduces ERF (Figure 3a) and then LES (Figure 3b), rather than following the actual sequence of NS-ERF data processing (first LES, then ERF).
> - Second, we directly prescribed time-dependent, heterogeneous sensible and latent heat flux grids as LES surface forcings. For this we utilized the pre-existing sensible and latent heat flux grids of Metzger et al. (2013). Metzger et al. (2013) coincidentally used ERF and observational data to derive these grids in their independent study, though any other appropriate dataset or methodology could have been used to provide the LES surface forcings. The Metzger et al. (2013) grids were merely a suitable dataset accessible to us at the time, irrespective of methodology.

Changes to the manuscript:

- Reversed the sequence of Figure 3a and Figure 3b to follow the actual sequence of NS-ERF data processing (here: first LES, then ERF).
- Removed any notion of ERF when citing the use of the Metzger et al. (2013) sensible and latent heat flux grids, to forego conflation with NS-ERF.
- In Sect. 2 Materials and methods add a paragraph that links the introduction to a "big picture" overview of the subsequent methods subsections. Distinguish between (i) the novel NS-ERF approach to OSSEs and (ii) its maiden, example application to the CHEESEHEAD19 case study.
- In the beginning of Sect. 2.2, added a flowchart of the generalized NS-ERF approach that acts as a "visual glossary" by inter-relating individual components/terminology (e.g., optimality criterion, hypothesis, objective, observing system design, scientific return etc.). Explained how these generalized components are then specifically realized in the CHEESEHEAD19 case study.
- Explained that the numerator in Eq. 1 varies according to the different OSDs, and that the denominator does not vary.

*After reading the results (L. 440) I now see that there is a baseline OSD for the tower-only dataset and 12 other OSDs for the combined tower-aircraft. The tower-only dataset has a given spatial coverage that can be improved by aircraft sampling, and the goal is to maximize this coverage by deciding on certain flight tracks. By applying different OSDs with different flight angles, it was found that the spatial coverage is maximized in a perpendicular flight track (25% improvement). In table 1, similar analyses are given for energy balance ratio, and spatial patterning, the other two optimality criteria.*

Author reply: Many thanks for this excellent summary! These relationships are also introduced in Sect. 2.3 l. 269ff.: "*Based on this super-sample we evaluated 13 candidate OSDs. Applying LES-ERF to 44 site-hours of data from the virtual EC tower network alone provided the* **baseline OSD**. *Combining data from the virtual EC tower network with one of the 3 flight sequences × 4 flight patterns = 12 airborne EC combinations provided 12* **alternative OSDs**. *Each of the alternative OSDs consisted of 44 site-hours virtual tower EC data and 11 flight tracks × 25 km = 275 km virtual airborne EC data. This configuration allows us to* **evaluate the change in the optimality criteria (Sect. 2.2) for each of the 12 joint tower and aircraft** **alternative OSDs relative to the tower-only baseline OSD**.".

Changes to the manuscript: Added the referee's "plain language summary" to above paragraph in Sect. 2.3.

*Reading section 2.4, I see that the LES are created with surface fields of H and LE created from the ERF. So, isn't this some circular reasoning? You are creating OSDs from ERFs that are benchmarked against LES created using ERF… Please explain*

Author reply: Please see our responses to above referee comment "*The methodology can be very confusing…*". We believe that the present comment originates from the same two ambiguities detailed in our response above.

Changes to the manuscript: Please see our changes in response to above referee comment "*The methodology can be very confusing…*".

*I think the results are clear and easy to follow but as mentioned previously the methodology can be a little confusing and I think needs more synthesis and needs to be clearer.*

Author reply: Please see our responses to above referee comment "*The methodology can be very confusing…*". We believe that the present comment originates from the same two ambiguities detailed in our response above.

Changes to the manuscript: Please see our changes in response to above referee comment "*The methodology can be very confusing…*".

*It is my understanding that the LES_ERF approach is designed specifically for the combination of airborne EC measurements with tower EC measurements. If this is the case this needs to be stated clearly in the abstract and perhaps even in the title.*

Author reply: Many thanks for sharing this observation! The presented approach to OSSEs is indeed widely applicable beyond combining airborne EC measurements with tower EC measurements. We feel that the impression of limited applicability likely originates from the current manuscript structure lacking separation between (i) the novel, general Numerical Simulation - Environmental Response Function (NS-ERF) approach to OSSEs and (ii) its maiden, example application to the CHEESEHEAD19 airborne case study. Furthermore, the manuscript does not currently specify which alternative form NS-ERF components could take for natural climate solutions, emission inventory validation, urban air quality, industry leak detection, and multi-species applications.

Changes to the manuscript:

- Re-organized manuscript to more clearly distinguish between (i) the novel NS-ERF approach to OSSEs and (ii) its maiden, example application to the CHEESEHEAD19 airborne case study.
- Reworded Sect. 2.2 header to "NS-ERF observing system simulation experiments". Added a flowchart/"visual glossary" to the beginning of the section for inter-relating generalized NS-ERF components/terminology.
- Substantiated extensibility claim by adding concrete examples for case study substitution where use cases are mentioned in the manuscript. In the discussion Sect. 4.2, spelled out how to extend to any sort of study where spatially and/or temporally sparse observations of a surface or atmospheric property X need to be combined with spatially and/or temporally more extensive observations of covariates Y to improve the spatial and/or temporal continuity of X.

*To me, it is not clear what the "scientific return" means. How can you double the scientific return? Isn't this all subjective? Whatever information you gained by using one flight path instead of the other depends on how you interpret it. Doesn't it? What the results show is that flight patterns with a track angle of 45-90 double the percent improvement based on three optimality criteria, when compared to parallel flight patterns. I'm also not sure how you can "order-of-magnitude improve flight operation and crew safety". The last two statements are part of the main conclusion but the way they are quantified seems subjective.*

Author reply:

- We agree with the referee's sentiment that "*Whatever information you gained by using one flight path instead of the other depends on how you interpret it*".
- OSSEs in general aim to provide an objective interpretation of "information gain" based on quantifying the ability to address the science objectives of a given application. To systematically realize this aim for surface-atmosphere field measurements we created the NS-ERF approach to OSSEs, and provide a case study example (CHEESEHEAD19 airborne design).
- In Sects. 2.2 and 2.3 we introduce the relevant NS-ERF components and their terminology in the text (e.g., optimality criterion, hypothesis, objective, observing system design, **scientific return** etc.). However, we do not currently provide a clear conceptual overview of the relationships among these components, thus creating a lack of context for objectively interpreting the CHEESEHEAD19 case study results.
- Sect. 4.1 specifies: "*On the other hand, the parsimonious number of only **6 flight sequences** and an even smaller number of 3 sets of waypoints simplify flight planning and navigation. In combination with entirely avoiding the town and airfield of Park Falls this **promotes flight crew safety by an order of magnitude compared to the originally envisioned 48 flight sequences**. Specifically, it frees up the flight crew from arduous navigation details, thus reducing fatigue, increasing awareness during the 100 m low-level flight maneuvers, and ultimately reducing the margin for human error.*"

Changes to the manuscript:

- Replaced "scientific return" with "information gain" throughout the manuscript.
- Reworded the title to "Novel approach to observing system simulation experiments improves information gain of surface-atmosphere field measurements".
- Reworded Sect. 2.2 header to "NS-ERF observing system simulation experiments". Added a flowchart/"visual glossary" to the beginning of Sect. 2.2 for inter-relating NS-ERF components/terminology including the aim to maximize information gain. At the end of the section (following "*We then used the arithmetic mean and standard deviation to aggregate CR1–CR3 across flight patterns, flight sequences, and ultimately among themselves into*

*a single score (Sect. 3.3).*"): defined information gain specifically for the CHEESEHEAD19 case study as the arithmetic mean of three optimality criteria that directly correspond to CHEESEHEAD19 science objectives.

- In Sect. 5 Conclusion, replaced *"order-of-magnitude improve flight operation and crew safety"* with "improve flight operation and crew safety by reducing the number of flight sequences from an originally envisioned 48 to a parsimonious number of 6".

*Lastly, a question of applicability. How many other researchers are in the capacity to apply airborne EC measurements with such large-scale deployment of towers and the capacity to run computationally expensive LES at this scale? The authors present a good analysis of other large-scale field campaigns in section 4.2 but still, I'm not sure about the use of this approach to support the last conclusion of the abstract that "the approach lends itself to optimize observing system designs also for natural climate solutions, emission inventory validation, urban air quality, industry leak detection, and multi-species applications" What would be the cost-benefit analysis of implementing a large-scale field campaign like this for every natural climate solution project?*

Author reply:

- Many thanks for raising this point. The presented NS-ERF approach to OSSEs is applicable to field measurements in general, and large-scale deployments or even an airborne component are by no means a requirement. For additional detail please see our responses to the above referee comment "*It is my understanding that the LES_ERF approach is designed specifically for the combination of airborne EC measurements with tower EC measurements...*"
- So when designing a natural climate solution (or other) project, NS-ERF could be applied at that project scale, e.g. much smaller compared to CHEESEHEAD19. Furthermore, an alternate NS-ERF configuration could be considered such as replacing LES with closure modelling or RANS as discussed in Sect. 4.3. Such modular adjustments and scalability reduce computational expense and makes the general NS-ERF approach to OSSEs accessible to a wide range of applications.
- The list of large-scale field campaigns in the manuscript are meant to provide context for the CHEESEHEAD19 airborne case study.

Changes to the manuscript:

- Please see our changes in response to above referee comment "*It is my understanding that the LES_ERF approach is designed specifically for the combination of airborne EC measurements with tower EC measurements...*"
- Specified that the NS-ERF approach to OSSEs is applicable to field measurements in general, and large-scale deployments or even an airborne component are by no means a requirement.
- Expanded concrete examples for case study substitution with scalability considerations incl. project size, NS-ERF configuration options, etc.

*31 This approach doubled the ability to explore energy balance closure? This is a very subjective statement. How do you double the ability to explore?*

Author reply: Thanks for this pointer. Please see our responses to above referee comment "*To me, it is not clear what the "scientific return" means…*". We believe that the present comment originates from the same ambiguity detailed in our response above.

Changes to the manuscript: Please see our changes in response to above referee comment "*To me, it is not clear what the "scientific return" means…*". Furthermore, changed "…*doubled CHEESEHEAD19's ability…*" to "doubled CHEESEHEAD19's potential" throughout the manuscript.

*371. Please specify the subsection in Section 4 where we can learn why the PBL was so low*

Author reply: Thanks for the pointer!

Changes to the manuscript: Added cross-reference to Sect. 4.1 where the issue is discussed in detail.

*409 Should this be "near-surface moisture"? "Land surface moisture" sounds like soil moisture and is redundant.*

Author reply: Thanks for this pointer.

Changes to the manuscript: Changed "land surface moisture (LSM)" to "near-surface moisture (NSM)" and "land surface temperature (LST)" to "near-surface temperature (NST)".

*L.465 Indicate in the table that these are percent values*

Author reply: Many thanks for your suggestion.

Changes to the manuscript: Updated Table 1 – Table 3 accordingly.

*I find that in the introduction and methods there are multiple statements that are hard to follow, such as:*

*192 "Here, we propose the extensible LES-ERF approach that explicitly simulates the joint scientific return in response to different candidate OSDs for addressing user-defined design hypotheses"*

*I think part of the problem is the use of the term "scientific return", which should be reevaluated.*

Author reply: Thanks for this pointer. Please see our responses to above referee comment "*To me, it is not clear what the "scientific return" means…*". We believe that

the present comment originates from the same ambiguity detailed in our response above.

Changes to the manuscript: Replaced "scientific return" with "information gain" throughout the manuscript. For additional detail please also see our changes in response to above referee comment "*To me, it is not clear what the "scientific return" means…*".

*I also think that an effort to synthesize the introduction and the methods would help the readability of the paper.*

Author reply: Agreed, many thanks for this observation!

Changes to the manuscript: In Sect. 2 Materials and methods added a paragraph that links the introduction to a "big picture" overview of the subsequent methods subsections. Distinguished between (i) the novel NS-ERF approach to OSSEs and (ii) its maiden, example application to the CHEESEHEAD19 case study in the form of LES-ERF.

*L.286 What is the LES time step?*

Author reply: The LES time step was set at 0.4 s for the analysis period. Mentioned in line 303.

*Technical comments*

*Should the verb be "creating" rather than "observing". Aren't you creating these OSSE?*

Author reply: We were not able to locate this comment in the manuscript.

**References**

Margairaz, F., Pardyjak, E. R., and Calaf, M.: Surface thermal heterogeneities and the atmospheric boundary layer: The relevance of dispersive fluxes, Boundary Layer Meteorol., 175, 369-395, doi:10.1007/s10546-020-00509-w, 2020.

Metzger, S., Xu, K., Desai, A. R., Taylor, J. R., Kljun, N., Schneider, D., Kampe, T., and Fox, A.: Spatio-temporal rectification of tower-based eddy-covariance flux measurements for consistently informing process-based models, 46[th] AGU annual Fall Meeting, San Francisco, U.S.A., 9 - 13 December, 2013.

Raupach, M. R., and Shaw, R. H.: Averaging procedures for flow within vegetation canopies, Boundary Layer Meteorol., 22, 79-90, doi:10.1007/BF00128057, 1982.

Sühring, M., Metzger, S., Xu, K., Durden, D., and Desai, A.: Trade-offs in flux disaggregation: a large-eddy simulation study, Boundary Layer Meteorol., 170, 69-93, doi:10.1007/s10546-018-0387-x, 2018.

Xu, K., Sühring, M., Metzger, S., Durden, D., and Desai, A. R.: Can data mining help eddy covariance see the landscape? A large-eddy simulation study, Boundary Layer Meteorol., Online First, doi:10.1007/s10546-020-00513-0, 2020.